# Topical phage therapy in a mouse model of *Cutibacterium acnes*-induced acne-like lesions

Amit Rimon [1,2], Chani Rakov[1], Vanda Lerer[1], Sivan Sheffer-Levi[3], Sivan Alkalay Oren[1], Tehila Shlomov[4], Lihi Shasha[1], Ruth Lubin [1], Khaled Zubeidat[1], Nora Jaber[1], Musa Mujahed[1], Asaf Wilensky[1], Shunit Coppenhagen-Glazer[1], Vered Molho-Pessach[3,5] & Ronen Hazan [1,5] ✉

Acne vulgaris is a common neutrophil-driven inflammatory skin disorder in which *Cutibacterium acnes* (*C. acnes*) is known to play a key role. For decades, antibiotics have been widely employed to treat acne vulgaris, inevitably resulting in increased bacterial antibiotic resistance. Phage therapy is a promising strategy to combat the growing challenge of antibiotic-resistant bacteria, utilizing viruses that specifically lyse bacteria. Herein, we explore the feasibility of phage therapy against *C. acnes*. Eight novel phages, isolated in our laboratory, and commonly used antibiotics eradicate 100% of clinically isolated *C. acnes* strains. Topical phage therapy in a *C. acnes*-induced acne-like lesions mouse model affords significantly superior clinical and histological scores. Moreover, the decrease in inflammatory response was reflected by the reduced expression of chemokine CXCL2, neutrophil infiltration, and other inflammatory cytokines when compared with the infected-untreated group. Overall, these findings indicate the potential of phage therapy for acne vulgaris as an additional tool to conventional antibiotics.

*Cutibacterium acnes* (*C. acnes*, previously termed *Propionibacterium acnes*) is a Gram-positive, lipophilic, anaerobic bacterium and a member of the skin microbiome[1]. *C. acnes* plays a pivotal role in the pathogenesis of acne vulgaris, a common chronic inflammatory disorder of the pilosebaceous unit[2], affecting 80% of the population during adolescence[3], as well as some adults[4]. Although *C. acnes* strains have been identified in association with healthy skin (phylotypes II and III)[5], other strains (phylotype IA) have been linked to the occurrence of acne[5]. The complex pathogenesis of acne involves androgen-mediated stimulation of sebaceous glands, follicular hyperkeratinization, dysbiosis within the pilosebaceous microbiome, and innate and cellular immune responses[1].

*C. acnes* activates the innate immune response to produce pro-inflammatory interleukin (IL)−1 by activating the nod-like receptor P3 (NLRP3) inflammasome in human sebocytes and monocytes[6]. Moreover, *C. acnes* can activate Toll-like receptor-2 in monocytes and trigger the secretion of the pro-inflammatory cytokines IL-12 and IL-8. IL-8 attracts neutrophils and leads to the release of lysosomal enzymes. These neutrophil-derived enzymes result in the rupture of the follicular epithelium and further aggravate inflammation[7].

For decades, acne has been treated with topical and oral antibiotics, including tetracycline (TET), doxycycline (DOX), minocycline (MC), erythromycin (EM), and clindamycin (CM), aimed at *C. acnes*[8]. Topical treatment regimens are preferred when possible, with level A

[1]Institute of Biomedical and Oral Research (IBOR), Faculty of Dental Medicine, The Hebrew University of Jerusalem, Jerusalem 91120, Israel. [2]Tzameret, The Military Track of Medicine, The Hebrew University-Hadassah Medical School, Jerusalem 91120, Israel. [3]Department of Dermatology, Hadassah Medical Center, The Faculty of Medicine, The Hebrew University of Jerusalem, Jerusalem 91120, Israel. [4]Department of Ophthalmology, Hadassah Medical Center, The Faculty of Medicine, The Hebrew University of Jerusalem, Jerusalem 91120, Israel. [5]These authors contributed equally: Vered Molho-Pessach, Ronen Hazan. ✉e-mail: ronenh@ekmd.huji.ac.il

recommendation strength[9], given the locally restricted effects and lack of systemic side effects[9]. However, an alarming global increase in antibiotic-resistant *C. acnes* strains has been documented over the last two decades[10]. For example, we examined the sensitivity profile of 36 clinical isolates of *C. acnes* from Israel to commonly used antibiotics, as listed above, and detected that the antibiotic resistance among this subset was 30.6% for at least one of the listed antibiotics[11]. These results correlate with the worldwide data of 20–60% resistant strains[11,12], indicating the need for alternative therapeutic strategies for treating acne vulgaris.

Bacteriophage (phage) therapy has been evolving as one of the most promising solutions for emerging antibiotic resistance[13]. Phages are bacterial viruses widely distributed in environments, replicating within bacteria and specifically killing their bacterial targets without harming other flora members. Therefore, they can be employed as living drugs for various bacterial infections, including acne[14].

Evidence of phage application against skin disease has been available since 1937[15]. The first *C. acnes* phage was described in 1964 by Brzin[16]. However, general interest in phage therapy has gained momentum in recent years, resulting in a renewed interest in developing this approach for treating acne vulgaris. This notion is reflected by the growing number of academic and industrial reports. Nevertheless, the research on this topic remains in its infancy. For instance, in-vivo phage therapy in an acne mouse model has been examined only by intralesional injections of *C. acnes* phages, and the efficacy of topical application remains unexplored[17–19].

Herein, we present the direct topical application of *C. acnes* phages in an in-vivo mouse model of acne vulgaris as proof of concept of phage therapy for acne. We examined our collection of *C. acnes* strains to establish their in-vitro susceptibility to eight novel phages isolated by our team. Using an acne mouse model, we evaluated the efficacy and safety of topical phage application in-vivo. To the best of our knowledge, this is the first report of direct topical phage application against *C. acnes*.

## Results

### Phage isolation

As part of routine work undertaken at the Israeli Phage Bank[20], potential phages targeting *C. acnes* were screened. Accordingly, we screened 49 acne vulgaris skin swabs and saliva samples (Table 1). On assessing this collection, eight phages targeting *C. acnes* were isolated (Table 2) and characterized as follows:

### Genome sequencing and analysis

The genome of each phage was fully sequenced and analyzed (Fig. 1a–c, Table 2, Supplemental Table S1). All phages were similar, with a genome size ranging between 29,535–30,034 bp (Table 2). To examine whether a phage is lysogenic or lytic, we performed a BLAST analysis and assessed for the presence of the whole phage genome sequence in bacterial genomes. A phage is considered lysogenic if most blast hits are within bacterial genomes. On the other hand, a phage is considered lytic if most of the blast hits are phages. In addition, we assessed for hallmarks of lysogens in phage genomes, such as Insertion Sequences (IS elements), repressors, and integrases. (Supplemental Table S2). We presumed that they were lytic phages without lysogeny abilities. The absence of repeat sequences indicated that their genomes exhibit a linear topology. Phylogenetic analysis revealed that the identified phages belong to the genus *Pahexavirus* of the *Siphoviridae* family. Although the phages were isolated separately, BLAST alignment of genomes revealed a high similarity between them (87–99%, with coverage of 95–98%, Fig. 1b, c). Moreover, these isolates are similar to genomes of numerous published *Cutibacterium acnes* phages, members of the *Siphoviridae* family[21].

Phages PAVL33 and PAVL34 were markedly distinct from the other six phages and close to each other (Fig. 1a–c). However, despite

### Table 1 | Patient characteristics

| Characteristics | Patients with acne (*n* = 36) No. of patients | % |
|---|---|---|
| Age | | |
| Median, years | 19 | – |
| Range, years | 11–30 | – |
| ≤20 years | 14 | 38.9 |
| >20 years | 22 | 61.1 |
| Gender | | |
| Male | 9 | 25.0 |
| Female | 27 | 75.0 |
| Acne severity[a] | | |
| Mild | 16 | 45.7 |
| Moderate | 15 | 42.9 |
| Severe | 4 | 11.4 |
| Acne duration[b] | | |
| 2 years or more | 28 | 82.4 |
| <2 years | 6 | 17.6 |
| Acne distribution | | |
| Only face | 20 | 55.6 |
| Face plus either back, chest, arms, or neck | 16 | 44.4 |
| Prior and current treatments[c] | | |
| No treatment | 6 | 18.8 |
| Treatment (antibiotic or other) | 26 | 81.2 |
| Antibiotic treatment (topical and oral) | 18 | 56.3 |

Several characteristics of the patients with acne vulgaris who were sampled for *Cutibacterium acnes* strains and phages. Source data is not provided as a Source Data file, to protect patients' identity.
[a]Information missing for 1 patient.
[b]Information missing for 3 patients.
[c]Information missing for 4 patients.

their high genome sequence identity, we considered them different, given that PAVL33 and PAVL34 were isolated from different samples. Compared with other phages, one difference between PAVL33 and PAVL34 was the coding sequence (CDS) QPB11516.1 (as termed in FD1 genome sequence, accession MW161461.1), present in FD1, FD2, FD3, PAVL20, PAVL 21, and PAVL 45 but not in PAVL33, PAVL34. This protein was annotated as a helix-turn-helix DNA-binding protein of *Siphoviridae* phages, deemed a regulator. Conversely, CDS QPB11846.1 (as termed in PAVL33 genome sequence, accession MW161466.1) was unique to PAVL33 and PAVL34 and was absent in the other six phages. This CDS was annotated as a hypothetical protein of the *Siphoviridae* phages (Fig. 1b and Supplemental Table S2).

One protein that did not follow this distribution was QPB11808.01 (as in PAVL33). This CDS was annotated as tape measure protein[22], present in PAVL20, PAVL33, PAVL34, PAVL45, and on the other four phages (Fig. 1b and Supplemental Table S2).

The genome of phages was free of known harmful virulence factors or antibiotic resistance elements, indicating their potential safety for phage therapy.

### Phage visualization

The geometric structure and morphological characteristics of the *C. acnes* phages were visualized using transmission electron microscopy (TEM) (Fig. 2a, b). All phages had a long non-contractile tail, attributed to the *Siphoviridae* morphology[23]. Given their genome similarities, we did not observe any differences. All phages exhibited a similar capsid geometrical structure to other members of their family[21], an

**Table 2 | Characterization of the isolated *Cutibacterium acnes* phages**

| Phage | Source of isolation | Genome size (bp) | %GC | GenBank accession | Tail length in nm (SD) | Capsid diameter in nm (SD) |
|---|---|---|---|---|---|---|
| FD1 | Saliva | 29,774 | 54.3% | MW161461.1 | 193.5 (16.5) | 65.5 (9.5) |
| FD2 | Saliva | 29,768 | 54.3% | MW161462.1 | 138.5 (8.5) | 52.1 (0.1) |
| FD3 | Saliva | 29,638 | 54.2% | MW161463.1 | 152.7 (9.3) | 54.3 (1.3) |
| PAVL20 | Skin | 29,800 | 54.3% | MW161464.1 | 138 (4.2) | 53.3 (7.7) |
| PAVL21 | Skin | 30,034 | 54.3% | MW161465.1 | 153.5 (3.5) | 54.65 (1.15) |
| PAVL45 | Skin | 29,772 | 54.3% | MW161468.1 | 136.2 (8.3) | 57.1 (1.9) |
| PAVL33 | Skin | 29,627 | 54.4% | MW161466.1 | 161.1 (1) | 53 (1.2) |
| PAVL34 | Skin | 29,535 | 54.3% | MW161467.1 | 156.1 (5.1) | 55.5 (2.4) |

Herein, eight new *C. acnes* phages were isolated and characterized. The measurements of the tail and capsid were performed using TEM images and are the average of two phages. The number in the names of the skin isolates PAVL relates to the patient sample number (See also Table 1). Source data are provided as a Source Data file.
*SD* standard deviation.

icosahedron, with a capsid diameter of 55.7 nm (±standard deviation = 3.9 nm; Fig. 2a and Supplemental Fig. S1), along with a 153.7 ± 17.4 nm long non-contractile tail (Fig. 2a and Supplemental Fig. S1).

## Host range coverage of phages in-vitro

Herein, we examined a collection of 36 clinical *C. acnes* isolates (Table 1), previously isolated by our team, and determined their sensitivity profile to the most frequently employed anti-*C. acnes* antibiotics (Table 3)[8]. Susceptibility to the described eight novel phages was assessed using a plaque assay on agar plates and in liquid cultures (Fig. 2c–e and Supplemental Fig. S2a–m). Of the 36, 32 (88%) *C. acnes* strains were sensitive to all eight phages, whereas the other four were resistant to all phages (Table 3). Interestingly, although the lysate originated from a single plaque, we observed various sizes of plaques on most plates (Fig. 2c and Supplemental Fig. S2a).

The 32 phage-sensitive strains included 11 strains resistant to at least one antibiotic, and 2 strains were resistant to all five antibiotics (Fig. 2d, e, Table 3, Supplemental Fig. S2a–e, and Table S3). Notably, the four phage-resistant strains were sensitive to all five antibiotics tested (Table 3). Thus, phages and antibiotics afforded 100% inhibition against all examined *C. acnes* strains (Supplemental Fig. S2e).

Validation of the plaque assays using liquid culture showed significant inhibition of bacterial growth with all phages (Fig. 2d, e and Supplemental Fig. S2f–m).

Accordingly, we hypothesized that antibiotic and phage sensitivity differences might correlate with the Single Locus Sequence Typing (SLST) type of the *C. acnes* strains; however, we failed to detect any such correlation (Supplemental Fig. S3a–c).

## Phage therapy in a *C. acnes*-induced acne-like lesions mouse model

We evaluated the potential of topical phage application for *C. acnes*-induced acne-like lesions using a mouse model. After screening several *C. acnes* strains, we arbitrarily selected strain 27 (Supplemental Table S3), a clinical isolate from a patient with severe acne vulgaris.

To select the phage for animal experimentation, phage stability in Carbopol gel (2.5%) was assessed by gel incubation and titer comparison to phage stability in Wilkins medium over time (Supplemental Fig. S4). Phages were incubated in an initial plaque-forming unit (PFU) of $6 \times 10^8 \frac{PFU}{mL} \left( \pm 7.55 \times 10^7 \frac{PFU}{mL} \right)$ at 4 °C for 30 days in gel or Wilkins media as a control (Supplemental Fig. S4a). No significant difference was observed between titers of phages incubated in a gel or Wilkins, which were both reduced in an approximate one log to an average PFU of $3.43 \times 10^7 \frac{PFU}{mL} \left( \pm 2.48 \times 10^7 \frac{PFU}{mL} \right)$, (*p*-value < 0.05). The most affected phage was PAVL33 $1.5 \times 10^7 \frac{PFU}{mL} \left( \pm 5 \times 10^6 \frac{PFU}{mL} \right)$. The most stable phage

was FD3, with a final PFU of $9 \times 10^7 \frac{PFU}{mL} \left( \pm 1 \times 10^7 \frac{PFU}{mL} \right)$, with no significant titer reduction during the 30-day incubation period (Supplemental Fig. S4a). The phage FD3 was selected for animal experimentation based on the stability results and its efficacy in both solid and liquid cultures (Supplemental Fig. S2a, H), which was comparable with the efficacy of a phage cocktail comprising FD1, FD3, and PAVL45 (Supplemental Fig. S4b). Based on these results, FD3 was selected as the phage for in-vivo assessment (Supplemental Fig. S2 and S4).

Accordingly, 38 8-week-old female ICR mice were administered two intradermal injections of strain 27, or saline for the Sham injection control group, to the dorsal surface on two consecutive days (days 1 and 2, respectively), followed by daily topical application of artificial human sebum[24] to the injection site (Fig. 3a). By day 3, bacterial injection yielded inflammatory acne-like lesions at the injection site, while the sham injection yielded slight skin elevation with a different morphology, which was self-limited. (Supplemental Table S4). Once *C. acnes*-induced lesions were established, mice were randomized into two groups (*n* = 17 per group). The treated group was treated with FD3 phage in Carbopol gel applied daily for five consecutive days; the infected-untreated group was treated with Carbopol gel only. Photographs of the lesions were obtained, and lesion diameter, elevation, and the presence of eschar were assessed daily. On day 10, the mice were euthanized, and tissue biopsies were performed and assessed histopathologically. Tissue biopsies were processed by homogenization, and the presence of bacteria and phages at the lesion site was determined using colony-forming unit (CFU) and PFU assays, respectively (Fig. 3b). All mice groups, including sham control and phage control groups did not exhibit any adverse events during this experiment.

## Phage and bacterial loads in acne-like lesions

Initially, we wanted to test the ability of FD3 phages to cross the full-thickness skin of mice ex-vivo. A transwell Franz diffusion cell system was established (Supplemental Fig. S5a). We found that the water-soluble Allora Red dye did not inhibit phage FD3 (Supplemental Fig. S5b), nor could it cross the mice's skin or parafilm control (Supplemental Fig. S5c). Additionally, FD3 crossed the skin of the mouse, as we could detect increasing PFU levels over time up to $9.33 \times 10^4 \pm 9 \times 10^4 \frac{PFU}{mL}$ after 6 hours in Carbopol gel, and a level of $4.7 \times 10^5 \pm 4 \times 10^5 \frac{PFU}{mL}$ after 3 hours when phages were placed in Wilkins medium (non-significant, Supplemental Fig. S5d–i).

Next, we quantified phage FD3 and *C. acnes* strain #27 from the lesion three days after the last administration of the phage-containing Carbopol gel by determining PFU and CFU counts, respectively, from homogenized 3 mm punched-skin biopsies. The

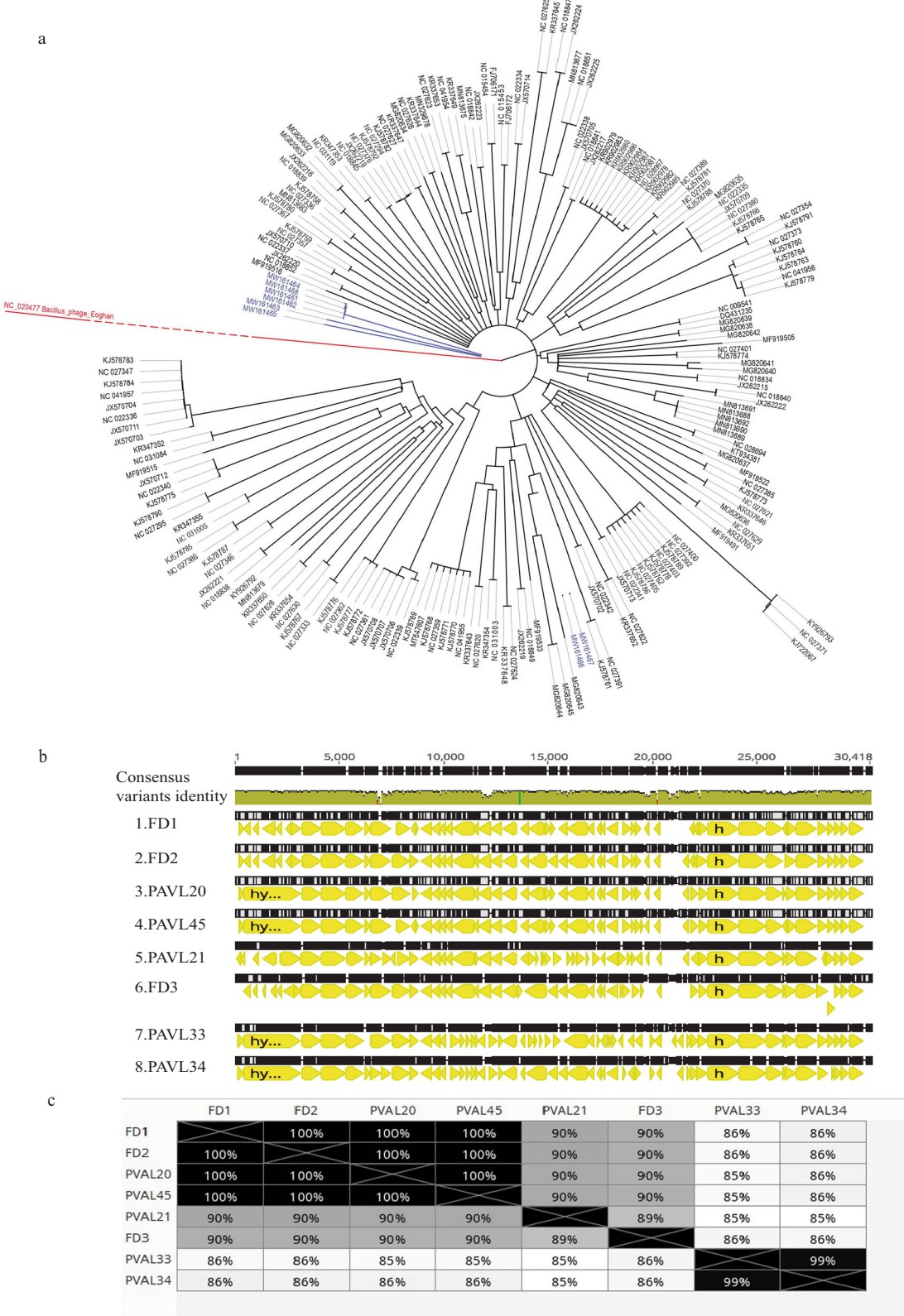

**Fig. 1 | Phage genomics. a** A phylogenetic tree of genomes of isolated *Cutibacterium acnes* phages (blue) and other *C. acnes* phages (black) from the GeneBank. The Bacillus phage Eoghan (accession: NC_020477) was employed as an outgroup. For more details and accession numbers, see Supplemental table S1. **b** Alignment of genomes of the eight isolated phages. **c** Distance table of the percentage of identity of phages. **b**, **c** were created using the Genious Prime 2022.2.1 package.

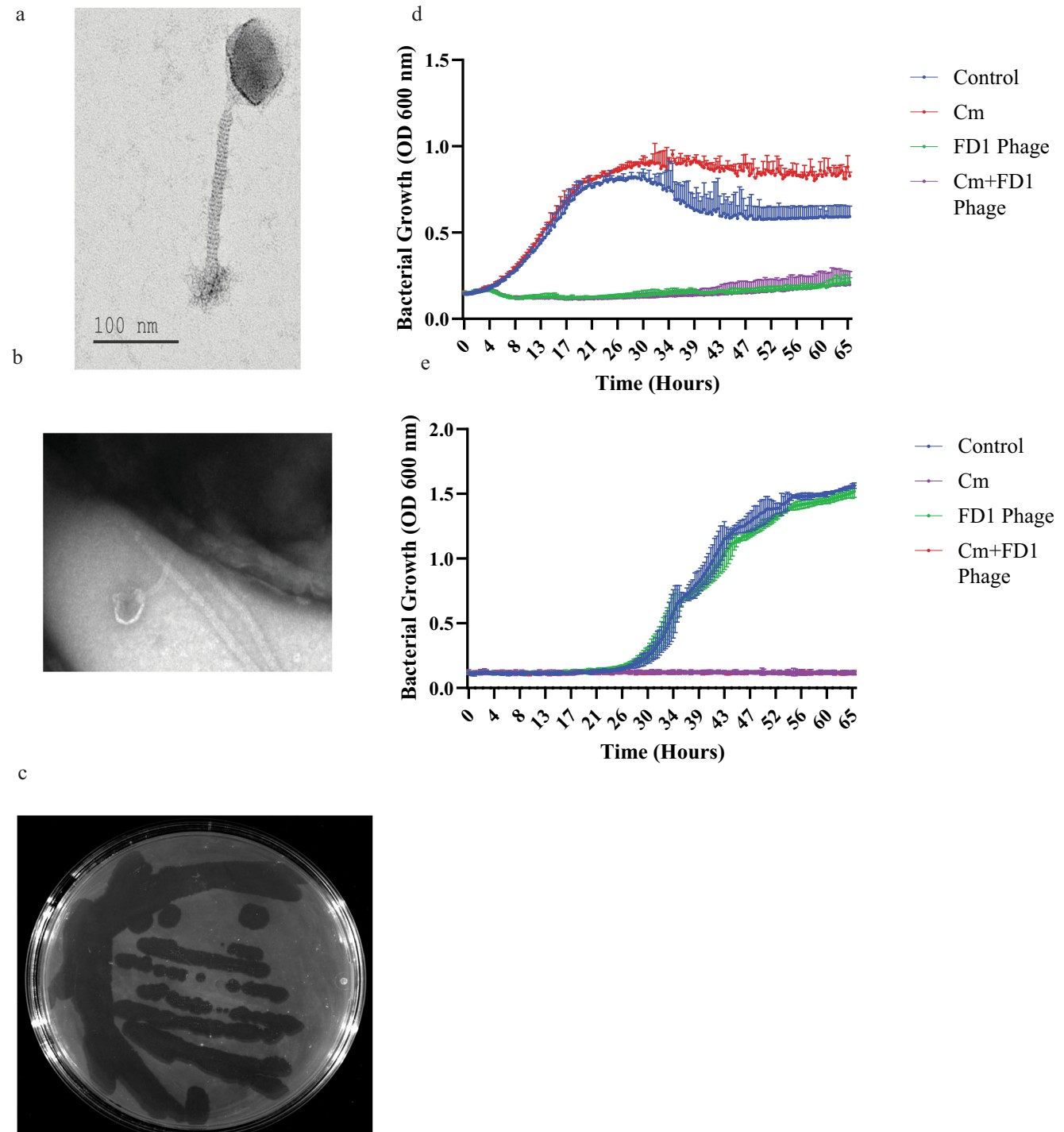

**Fig. 2 | In-vitro phage efficacy and morphology. a, b** Transmission electron microscopy of phage FD1 and FD1 bound to its bacterial target (**b**). In addition, images of all phages are presented in Supplemental Fig. S2a. This experiment was done twice (**c**) FD1 plaques on *Cutibacterium acnes* strain 27. Note the diverse sizes of plaques, although they originate from a single plaque. **d** The CM-resistant strain 28, growth with CM, phage FD1, and combination. The results are the average of triplicates, presented as mean ± standard deviation (SD). **e** FD1-resistant strain 21 growth with CM, FD1 phage, and their combination. The two lower curves, CM and CM + FD1, overlap. The results are the average of triplicates, presented as mean ± SD. CM, clindamycin. Source data are provided as a Source Data file.

average PFU of the treated group was $1.4 \times 10^4 \pm 1 \times 10^3 \frac{PFU}{mL}$ and as expected, no plaques were detected in the control groups ($p$-value < 0.01, Fig. 3b).

Regarding the bacteria, none was isolated from the sham-injected group ($p$-value < 0.05, Fig. 3b). The $p$-value of differences between the treated and infected-untreated groups was 0.09 (Fig. 3b). However, if the two outliers were excluded, the spontaneously recovered mice in the infected-untreated control group and one mouse that was not affected by the treatment, the average of the control group was $4.56 \times 10^4 \pm 2.9 \times 10^4 \frac{CFU}{mL}$ and that of the treated group was $6 \times 10^2 \pm 8 \times 10^2 \frac{CFU}{mL}$ ($p$-value < 0.01, Fig. 3b).

These results suggest that the FD3 phages could cross the full thickness of mice's skin and remain effective in the lesions (Fig. 3b and Supplemental Fig. S5).

**Table 3 | Phage, antibiotic, and combination sensitivity profile of the isolated *Cutibacterium acnes* strains**

| # | Any of the phages[a] | EM | EM + Phages[a] | CM | CM + Phages[a] | TET | TET + Phages[a] | DOX | DOX + Phages[a] | MC | MC + Phages[a] | All antibiotics | All antibiotics + Phages[a] |
|---|---|---|---|---|---|---|---|---|---|---|---|---|---|
| 1 | S | S | S | S | S | S | S | S | S | S | S | S | S |
| 2 | S | R | S | R | S | — | S | R | S | — | S | S | S |
| 4 | R | S | S | S | S | S | S | S | S | S | S | S | S |
| 5 | S | R | S | R | S | R | S | R | S | S | S | S | S |
| 6 | S | S | S | S | S | S | S | S | S | S | S | S | S |
| 7 | S | R | S | — | S | — | S | — | S | S | S | S | S |
| 8 | S | S | S | S | S | S | S | S | S | S | S | S | S |
| 9 | S | S | S | S | S | S | S | S | S | S | S | S | S |
| 10 | S | S | S | S | S | S | S | S | S | R | R | S | S |
| 11 | S | S | S | S | S | S | S | S | S | S | S | S | S |
| 13 | S | S | S | S | S | — | S | S | S | S | S | S | S |
| 14 | S | R | S | R | S | R | S | R | S | R | S | R | S |
| 15 | S | S | S | S | S | S | S | S | S | S | S | S | S |
| 18 | S | S | S | S | S | S | S | S | S | S | S | S | S |
| 19 | S | S | S | S | S | S | S | S | S | S | S | S | S |
| 21 | R | S | S | S | S | S | S | S | S | S | S | S | S |
| 22 | S | S | S | S | S | S | S | S | S | S | S | S | S |
| 23 | S | S | S | S | S | S | S | S | S | R | S | S | S |
| 24 | S | R | S | — | S | S | S | S | S | S | S | S | S |
| 25 | S | R | S | — | S | — | S | R | S | — | S | — | S |
| 27 | S | S | S | S | S | S | S | S | S | S | S | S | S |
| 28 | S | R | S | R | S | — | S | R | S | S | S | S | S |
| 30 | S | S | S | S | S | S | S | S | S | S | S | S | S |
| 31 | S | R | S | R | S | — | S | R | S | — | S | — | S |
| 32 | S | S | S | S | S | S | S | S | S | S | S | S | S |
| 33 | R | S | S | S | S | S | S | S | S | S | S | S | S |
| 35 | S | R | S | R | S | R | S | R | S | R | S | R | S |
| 36 | S | S | S | S | S | S | S | S | S | S | S | S | S |
| 37 | S | S | S | S | S | S | S | S | S | S | S | S | S |
| 40 | S | S | S | S | S | S | S | S | S | S | S | S | S |
| 41 | S | S | S | S | S | S | S | S | S | S | S | S | S |
| 43 | S | S | S | S | S | S | S | S | S | S | S | S | S |
| 44 | S | S | S | S | S | S | S | S | S | S | S | S | S |
| 47 | S | S | S | S | S | S | S | S | S | S | S | S | S |
| 48 | R | R | S | S | S | S | S | S | S | S | S | S | S |
| 49 | S | S | S | S | S | S | S | S | S | S | S | S | S |

The sensitivity of the C. acnes strains to phages was tested using plaque assays, and their antibiotic sensitivity profile was determined as previously described[11]. The antibiotics tested were erythromycin (EM), clindamycin (CM), tetracycline (TET), doxycycline (DOX), and minocycline (MC). The sensitivity is defined as sensitive (S), intermediate (I), or resistant (R). The inconsistency in isolate number is due to the exclusion of some isolates upon 16 S analysis, revealing that they are not C. acnes strains. For more details, see Supplemental Table S3.

[a]Given the absence of differences between phages and all tested C. acnes strains were either sensitive (S) or resistant to all tested phages (FD1, FD2, FD3, PAVL20, PAVL21, PAVL33, PAVL34, PAVL45).

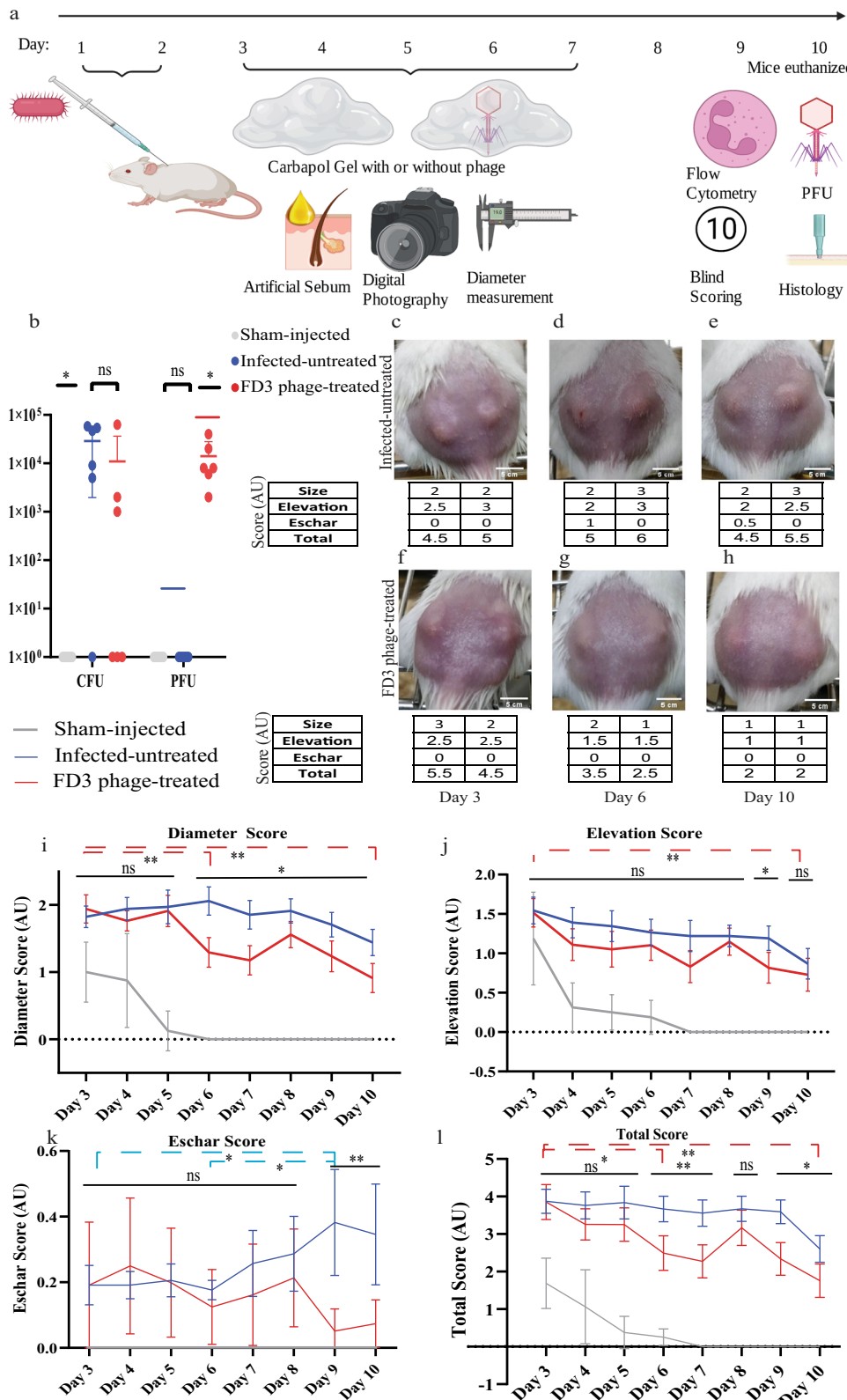

## Assessment of *C. acnes*-induced lesions in mice

We next employed the clinical and histological scores to assess the severity of the *C. acnes*-induced lesions (Supplemental Fig. S6–7). The scoring method employed in the present study was based on the acne-Leeds-grading system, along with other described scoring methods[24,25]. Based on the scores, daily application of FD3 for five consecutive days significantly improved the severity of the lesions, as reflected by their diameter, papulation/elevation, and presence and severity of eschar over the lesions, compared with features of sham-injected mice, and infected mice treated with vehicle only (Fig. 3c–h and Supplemental Table S4).

The first parameter was the difference in lesion diameters (Supplemental Fig. S6). The sham-injected mice score was 1 ± 0.53 arbitrary units (AU) (diameter of 2.4 mm) on day 3; it reduced to a score of 0 AU

**Fig. 3 | In-vivo phage therapy of *Cutibacterium acnes*-induced acne-like lesions.** **a** Schematic representation of the experiment. Mice were injected intradermally with *C. acnes* on days one and two, and artificial sebum, known to enhance the pathogenicity, was applied daily from day 1. Throughout days 3–7, mice were treated topically with Carbopol gel (2.5%) with or without FD3 phages. On day 10, the mice were euthanized, and their lesions were analyzed. Mice were followed and scored upon group allocation at day 3 until day 10 (This figure was prepared using biorender.com). **b** (Colony-forming unit (CFU) and plaque-forming units (PFU) counts from mice lesions. *n* = 6 lesions. *p*-value of the infected-untreated group in comparison to phage-treated group = 0.26, 0.03 for CFU, PFU accordingly. **c**–**h** Representative photographs of *C. acnes*-induced lesions in the infected-untreated mouse # 11 (**c**–**e**) and the treated mouse #12 (**f**–**h**) were obtained on days 3, 6, and 10, and their clinical scores. (Photos of the lesions of all mice, including the

sham-injected control group, can be seen in Supplemental Table S4). **i**–**l** Scores of diameter, *p*-value of the infected-untreated group in comparison to phage-treated group = 0.03 (**i**), degree of elevation, *p*-value of the infected-untreated group in comparison to phage-treated group = 0.07 (**j**), eschar, *p*-value of the infected-untreated group in comparison to phage-treated group = 0.02 (**k**), and combined scores, *p*-value of the infected-untreated group in comparison to phage-treated group = 0.01 (**l**) of the sham-injected control (gray), infected-untreated control (blue), and the treated (red) groups during the experimental period. *n* = 34 lesions. Data are presented as a mean ± 95% confidence interval. Student's *t* test two-tailed unpaired was used, **p*-value < 0.05, and ***p*-value < 0.001. Significances are presented between the FD3-treated group and the infected-untreated group on each day (straight lines) and inter-day comparisons (dashed lines). Source data are provided as a Source Data file.

by day 6, and was lower than the two infected groups throughout the whole experiment (*p*-value < 0.05, Fig. 3i). In the two infected groups, lesion diameters differed significantly from day 6 until 10. On day six, the scores were 2.09 ± 0.6 AU (diameter of 5.2 mm) in the infected-untreated group and 1.27 ± 0.642 AU (diameter of 3.6 mm) in the treated group (*p*-value < 0.001, Fig. 3i). The treated group exhibited a continuous reduction in lesion diameter, from an average score of 1.93 ± 0.6 AU (diameter of 4.6 mm on day 1) to 1.00 ± 0.654 AU (diameter of 2.9 mm on day 10; *p*-value < 0.001, Fig. 3i). Significant differences were also observed between days 9, 10, and from day 3 to day 5 (*p*-value < 0.05, Fig. 3i). In contrast, the average lesion diameter of the infected-untreated group was altered in a non-continuous manner, increasing from an average score of 1.82 AU ± 0.459 (diameter of 4.5 mm on day 1) to 2.09 AU ± 0.6 (diameter of 5.2 mm on day 4) during the early days of the experiment, subsequently decreasing to 1.7 ± 0.524 AU (diameter of 4.5 mm on day 10) at the end of the experiment (Fig. 3i). Moreover, we noted spontaneous reduction in lesion's diameter in the infected-untreated group toward the end of the experimental period (Fig. 3i). Overall, the mean of the infected-untreated group was 1.83 ± 0.19 AU, and that of the treated group was 1.47 ± 0.37 AU (*p*-value < 0.05, Fig. 3i).

Accordingly, we noted a significant difference in diameter scores between groups.

Next, we assessed the degree of elevation (papulation) as the second clinical parameter of acne-induced lesions (Supplemental Fig. S6). On day 3, the average elevation score of lesions in the sham-injected group was 1.18 ± 0.704 AU (non-significant, Fig. 3j), which reduced to 0 by day 7 and was significantly smaller than the two infected groups throughout days 4–10 (*p*-value < 0.05, Fig. 3j). On day 3, the average elevation score of lesions was 1.57 ± 0.486 AU and 1.52 ± 0.515 AU in the infected-untreated and treated groups, respectively (non-significant, Fig. 3j). However, on day 7, this parameter decreased to 0.83 ± 0.583 AU in the treated group; the infected-untreated group presented a value of 1.22 ± 0.567 AU (*p*-value < 0.005, Fig. 3j). On day 9, the average elevation score decreased to 0.816 ± 0.558 AU in the treated group and was 1.19 ± 0.444 AU in the infected-untreated group (*p*-value < 0.005, Fig. 3j). Overall, the mean of the infected-untreated group was 1.25 AU ± 0.19, whereas that of the treated group was 1.03 ± 0.25 AU (*p*-value = 0.07, Fig. 3j). However, this value was reduced in the treated group during the final experimental days (Fig. 3j).

In summary, we noted significant differences in the degree of elevation between lesions of the treated and infected-untreated groups.

We then examined a third clinical parameter, the presence of eschar in the *C. acnes*-induced lesions (Supplemental Fig. S6). While in the sham-injected group, no eschar was observed, the overall average eschar score for the infected-untreated group was 0.25 ± 0.077 AU versus 0.16 ± 0.069 AU in the treated group (*p*-value < 0.05, Fig. 3k). On analyzing each day separately, the treated group presented significantly reduced average eschar scores on days 9 and 10. On day nine, the average eschar score of the treated group was 0.051 ± 0.192 AU,

while that of the infected-untreated group was 0.382 ± 0.464 AU. On day 10, the average score of the treated group was 0.074 ± 0.209 AU, while that of the infected-untreated group was 0.346 ± 0.441 AU (*p*-value < 0.001 for both days 9 and 10, Fig. 3k).

The average eschar score of the infected-untreated control group differed between days 9, 3, and 6 (*p*-value < 0.05, Fig. 3k).

In summary, we noted a significant reduction in eschar levels in the treated group compared to the infected-untreated control group.

To simplify the scoring method and assess the lesion with a single score, we compared a combined score of the three clinical parameters mentioned above (termed the "Total Clinical Score") (Supplemental Fig. S6). Using this score, we noted that differences between infected-untreated and treated groups were significant on days 6, 7, 9, and 10 (*p*-value < 0.05 at any time point, Fig. 3l). In the infected-untreated group, the score reduced from 3.368 ± 0.746 AU on day 3 to 3.127 ± 0.767 AU on day 10 (non-significant, Fig. 3l). In the treated group, the score was significantly reduced from 3.39 ± 1.010 AU on day 3 to 2.051 ± 0.1020 AU on day 10 (*p*-value < 0.001, Fig. 3l). The sham-injected group score was lower than the two infected groups throughout the experiment (*p*-value < 0.001, Fig. 3l).

In summary, the Total Clinical Score, which represents the overall severity of the lesions, increased significantly in the infected-untreated group compared with that in the treated group (Fig. 3l and Supplemental Fig. S6).

## Histopathological evaluation of *C. acnes*-induced lesions

On day 10 of the experiment, biopsies were obtained from all groups, and cellular infiltration of lesions was assessed by histopathology (Fig. 4). The skin biopsies of the sham-injected mice had no significant pathological findings and were within the normal skin range (Fig. 4a–c and Supplemental Fig. S7). The infected-untreated mice showed a markedly severe and acute pyogranulomatous inflammatory process when compared with the treated mice. In some cases, inflammation involved the dermis and subcutaneous tissue (Fig. 4d). A nodular infiltrate was observed, with an area of necrosis (Fig. 4e, f. In contrast, inflammation was less severe in the treated group, primarily exhibiting chronic granulomatous infiltration, involving only subcutaneous tissue below the panniculus carnosus, mixed with minimal neutrophils (Fig. 4g–i).

As there was no detectable pathology in the sham-injected group, it was not scored. The total histological score was calculated to compare the two infected groups. Accordingly, we assessed four lesions from each group using two different parameters: the area of the lesions and the histological score (Supplemental Fig. S7). The average area of the lesion was 1.783 ± 1.07 mm² and 0.6458 ± 0.073 mm² in the infected-untreated and treated groups, respectively (non-significant, Fig. 4g and Supplemental Fig. S7). The average histological score was 2.25 ± 0.43 AU and 1.25 ± 0.43 AU in the infected-untreated and treated groups, respectively (*p*-value < 0.05, Fig. 4g and Supplemental Fig. S7). The Total Histological score was obtained by combining the histological score and lesion's area. It equaled 4.033 ± 1.45 AU and 1.896 ± 0.4196 AU in the infected-untreated and treated groups,

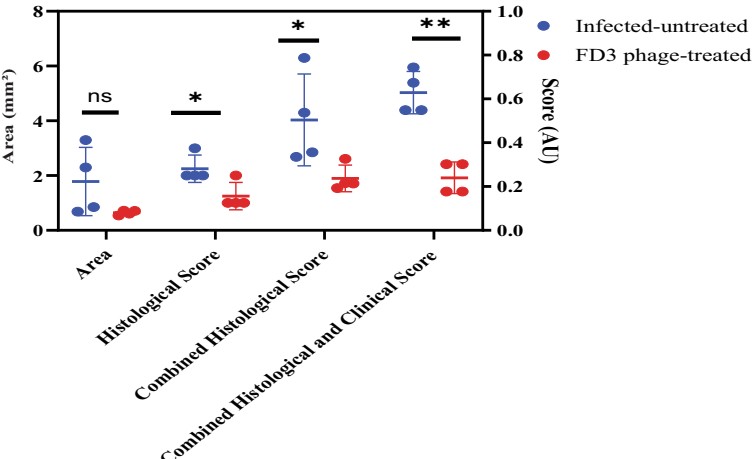

Sham-injected Infected-untreated FD3 phage-treated

respectively (*p*-value < 0.05, Fig. 4g and Supplemental Fig. S7). In summary, treated mice exhibited a reduced histological score compared to the infected-untreated control mice, while the biopsy of the sham-injected mice's histology was within the normal range of healthy skin (Fig. 4 and Supplemental Fig. S7).

The combination of the Total Clinical Score and the Total Histological score was 4.579 and 2.471 AU in the infected-untreated and treated groups, respectively (*p*-value < 0.001, Figs. 3l and 4g, Supplemental Fig S7, and Table S4).

## Evaluation of the inflammatory response

Given that acne is an immune-driven process critically mediated by macrophages and neutrophils[1,7,26], we examined the inflammatory immune response and neutrophilic infiltrate in biopsies collected at

**Fig. 4 | Histology of the *Cutibacterium acnes*-induced lesions.** Mice were euthanized on day 10. **a–c** sham-injected control group,at magnification of (×4, ×10, ×20, A-C accordingly). The skin is within normal range and therefore not scored. **d** Nodular inflammation (arrow) involving the dermis and subcutaneous tissue at ×4 magnification in the infected-untreated group scored 2. **e** Nodular infiltrates with an area of necrosis with cellular infiltration (arrowhead) at ×10 magnification in the infected-untreated group. **f** ×20 magnification of the infected-untreated group. **g** Nodular inflammation (arrow) involving only the subcutaneous tissue below the panniculus carnosus at ×4 magnification in the treated group scored 1. **h** A nodular cellular infiltration (arrowhead), numerous vacuolated cytoplasm mixed with a minimal number of neutrophils at ×10 magnification of the treated group. **i** ×20 magnification of the treated group This experiment was done twice. **j** Affected lesion area was measured, and scored; for further details, see Fig. 7 and Supplemental Fig. S4, *p*-value of the infected-untreated group in comparison to phage-treated group = 0.11, 0.03, 0.04, <0.001 for Area, Histological Score, Combined Histological Score, and Combined Histological and Clinical Score accordingly. *n* = 4 mice. Data are presented as mean ± standard deviation (SD). Student's *t* test two-tailed unpaired was used, **p*-value < 0.05, and ***p*-value < 0.001. Source data are provided as a Source Data file.

various time points using four methods: **a.** Immunofluorescence staining of biopsies for Ly6G, a neutrophilic marker[27] (Fig. 5a–e); **b.** Flow cytometric of several markers (Fig. 6a–e); **c.** Real-time PCR to assess the expression of various cytokines and chemokine genes Fig. 6f); ELISA to assess the expression levels of *IL1-β* (Fig. 6g), Accordingly, we assessed four groups of mice: **a.** sham-injected **b.** phage control, **c.** Infected-untreated, and **d.** infected and phage-treated (treated) at three time points: days 3, 5, and 7. Each group comprised 3–5 animals (Table 4).

Neutrophils were quantified by assessing the immunofluorescence of Ly6G$^+$ from an average of 5 fields, with images obtained from each group at the same magnitude (20×, Fig. 5a–d). The average number of neutrophils per field in the sham-injected untreated group was 2.6 ± 1.14, and phage control was 2.8 ± 1.64 (non-significant *vs.* Sham injection untreated, Fig. 5a, b, e). Average number of neutrophils per field in the treated mice was 2.6 ± 2.07 (non-significant *vs.* both sham injection groups, Figs. 5a, d, e) and 27.6 ± 5.59 in the infected untreated-group (*p*-value *vs.* all other groups <0.001, Fig. 5a–e). All images can be found in the BioStudies data repository (https://www.ebi.ac.uk/biostudies/studies/S-BSST889).

In addition, flow cytometry was used to analyze various markers that characterize immune cell populations, mainly macrophages and neutrophils (Fig. 6a–e and Supplemental Fig. S8). In the skin specimen of the sham-injected control group the average percentage of neutrophils (%NT) was 6.137% ± 2.07% (Fig. 6a, d) without any significant alteration on days 5 and 7 with or without phage application (Fig. 6d). In the infected groups, %NT was 55.97% ± 19.89% (Fig. 6b, d) on day 3 (before phage treatment). On day 7, the average %NT was 43.08% ± 25.26% and 18.48% ± 10.26%in the untreated and treated groups, respectively, with significant differences noted between days 3 and 7 in the treated group (*p*-value = 0.02, Fig. 6c, d), as well as between the phage-untreated and treated groups over the experimental period (*p*-value = 0.02, Fig. 6d).

The %NT of treated mice on days 5 and 7 was non-significant compared with the sham-injected control, indicating their %NT reduced to the basal level observed in the sham-injected control (*p*-value = 0.07, 0.09 accordingly). The %NT in uninfected-untreated mice was not reduced to the uninfected control basal levels (*p*-value < 0.05; Fig. 6d, e). Moreover, we detected a significant increment in CD64$^+$ macrophages only in the *infected*, phage-treated group on day 7 in comparison to all other groups (47.22% ± 5.81%, *p*-value < 0.001, and on day 5 in comparison to the sham-injected control (18.33% ± 1.8%, *p*-value < 0.05, Supplemental Fig. S8a).

To further validate the results of inflammation, the expression levels of several hallmark genes of inflammation were determined using the gene of the rRNA 18S as the housekeeping gene (Fig. 6f). Gene expression levels of *IL17* were significantly reduced in the treated group when compared with the infected-untreated group (*p*-value = 0.03), approaching levels observed in the sham-injected, phage control groups (non-significant, Fig. 6f). In addition, gene expression levels of *CXCL2*, *TNF-α*, and *IL-6* in the treated group were significantly reduced in comparison to the infected-untreated group (*p*-value < 0.05, <0.001 and <0.001, respectively), but not to control groups levels (*p*-value < 0.05, <0.001, and <0.05 respectively, Fig. 6f).

Finally, we performed ELISA to measure the levels of the excreted *IL1-β* in the blood, a cytokine known to mediate neutrophil migration[28] (Fig. 6g). As detected by the ELISA results, the mean values of the Sham injected and phage control were 142.9 ± 41.56 ρg/ml, 108.4 ± 34.41 ρg/ml accordingly (non-significant, Fig. 6g). In the infected-untreated group *IL1-β* levels were 711.8 ± 92.17 ρg/ml *p*-value < 0.001 compared with both control groups, Fig. 6g). In FD3-treated group *IL1-β* level was 462.4 ± 86.93 ρg/ml (*p*-value < 0.05 compared with the infected-untreated group, Fig. 6g),

In summary, these results reveal that phage treatment reduce *C. acnes*-induced inflammation, as evidenced by the decreased neutrophil migration to the lesion, reduced expression of inflammatory cytokines and chemokines in the lesions, and excretion of *IL1-β* into the blood. (Figs. 5 and 6a–g).

## Discussion

In the present study, we aimed to evaluate the potential of phage therapy in a model of *C. acnes*-induced acne-like lesions. Accordingly, we isolated eight new phages and tested their efficacy in-vitro against a collection of clinical *C. acnes* isolates, whose antibiotic-resistant profile was previously determined by our group[11]. The isolated phages were found to be similar both genetically and phenotypically. Distinguishing their efficacy against bacteria, i.e., an individual strain was either sensitive or resistant to all phages, was challenging. Nevertheless, resistant strains (but not only) were found to be sensitive to commonly used antibiotics; thus, phages and antibiotics provided 100% inhibitory coverage against our *C. acnes* collection. Next, we selected the most virulent bacteria to induce lesions mimicking acne vulgaris, followed by treatment with a selected phage. The examined phage treatment could reduce the bacterial load and inflammation in induced lesions.

Herein, our findings demonstrated the potential of phage therapy as an additional tool to antibiotic treatments, given the growing antibiotic resistance. Moreover, the examined clindamycin-resistant strains grew in the presence of the antibiotic while exhibiting superior growth in the absence of antibiotics (Fig. 2d), perhaps because the antibiotic inhibits certain programmed-cell death or autolysis mechanisms induced by *C. acnes*[29,30]. If accurate, this observation further warrants the need for additional anti-bacterial weapons in our current arsenal.

The *C. acnes* cohort was dominated by SLST type A1 (phylotype IA1 in the traditional typing method)[31], which is considered acne-associated[32]. The 88.8% antibiotic-sensitive strains in our cohort were collected in Israel[11] (Fig. 2, Table 3, Supplemental Fig. S2, and Table S3) and agreed with other worldwide reports, with a rate of approximately 88% antibiotic sensitivity among clinical isolates[21,22,33–36]. Furthermore, although we noted a significant correlation in our *C. acnes* collection between the A1 SLST type and the antibiotic resistance strains, no correlation was detected between any SLST type and phage-resistance strains (Supplemental Fig. S3). This finding could be attributed to the small sample of strains or indicate a higher prevalence of antibiotic-resistant strains than phage-resistant ones among clinical strains, owing to the common antibiotic treatments.

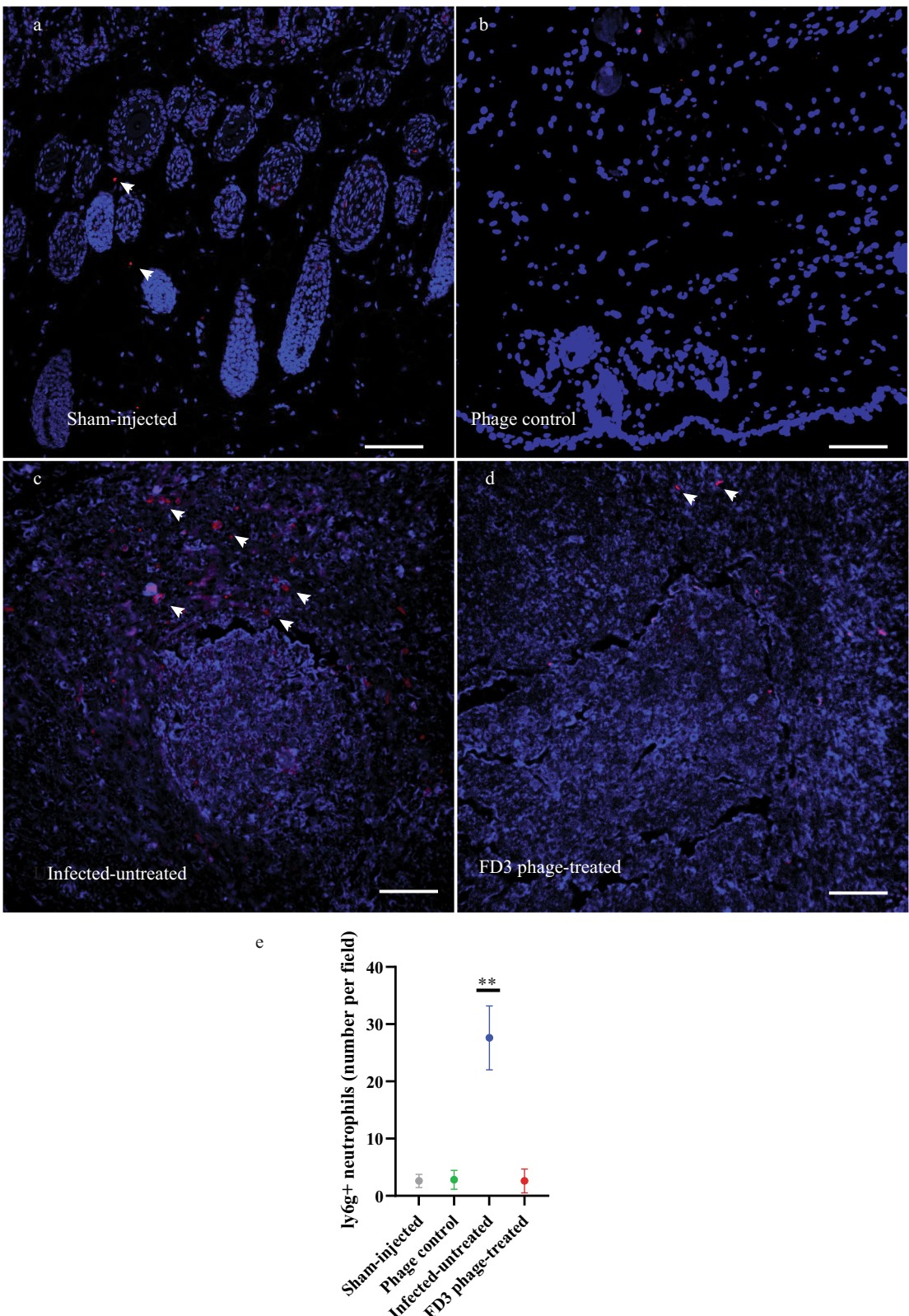

**Fig. 5 | Immunofluorescence staining of Ly6G + neutrophils. a**–**c** anti-Ly6G (red) and DAPI (blue) were imaged with a fluorescence microscope at ×20 magnite. Some neutrophils are denoted with an arrow **a** Sham-injected group, **b** Phage control, **c** untreated group, **d** FD3-treated group, and **e** numbering Ly6G+ neutrophils. Scale bar: 200 μm. The neutrophils were counted in 5 fields (20×) per slice.

All images can be found in the data repository https://www.ebi.ac.uk/biostudies/ studies/S-BSST889. *p*-value of the infected-untreated group in comparison to all other groups <0.0001, *n* = 5 fields, this experiment was done twice. Data are presented as mean ± standard deviation (SD). Student's *t* test two-tailed unpaired was used, **\*\*p*-value < 0.001. Source data are provided as a Source Data file.

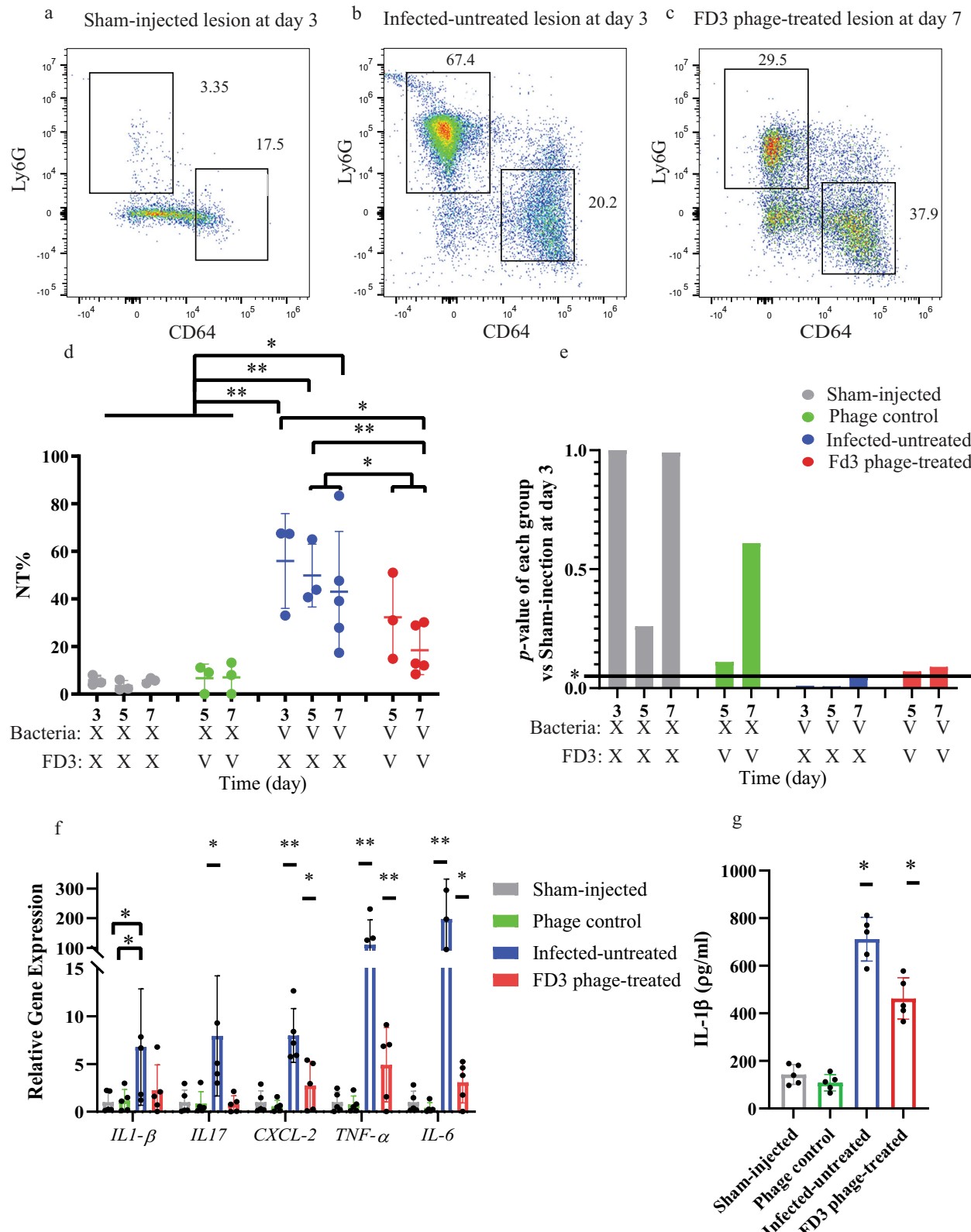

Interestingly, all PAVL phages were isolated from skin swabs that simultaneously yielded phage-resistant bacteria (Supplemental Table S3). This observation demonstrates the complex predator-prey co-existence of phages in the presence of both sensitive, allowing phage growth, and resistant bacterial strains, without eliminating any of them. Further studies are needed to validate this observation and test the nature of the resistance, e.g., mutations or acquisition of

anti-phage defense mechanisms[37], abortive system[38], or activation of adaptive systems such as the Type II CRISPR-Cas system that was found in *C. acnes*[39]. This co-existence and the presence of defense mechanisms highly impact phage therapy design and treatments. However, in contrast to antibiotics, several steps can be taken when resistance to phage therapy emerges, including the isolation of new phages from nature, cycles of evolutionary adaption ("phage training")[40], and phage

**Fig. 6 | Inflammatory response in *Cutibacterium acnes*-induced lesions following phage application.** Mice group details can be found in Table 4. On day 3, mice were allocated to treatment or vehicle groups; therefore, there were no treated mice at this time point. **a–c** Representative flow cytometry analysis markers of neutrophils (Ly6G) and macrophages (CD64). **a** A lesion from the Sham-injected control group on day 3. **b** A lesion from the Infected untreated group on day 3. **c** A lesion from the FD3 phage-treated group on day 7. **d** Quantification of the neutrophils' number in flow cytometric analysis, a *p*-value of the infected-untreated group in comparison to the phage-treated group, in days 5 and 7 combined = 0.02, *n* = between 3–5 mice as described in Table 4. Note that the macrophage results are presented in Supplemental Fig. S8a. **e** the *p*-values of each group compared with the uninfected control on day 3. The horizontal line represents a *p*-value of 0.05.

Infected-untreated group *p*-value = 0.04 while the phage-treated group 0.09 indicates phage-treated group normalization. **f** Expression of immune-relevant genes determined by quantitative PCR in sham-injected, phage control, infected-untreated, and phage FD3-treated. *rRNA* 18S was employed as the housekeeping gene, and the sham-injected group was the normalizer group., *p*-value of the infected-untreated group in comparison to phage-treated group = 0.16, 0.03, 0.01, <0.001, <0.001 for *IL-1β, IL-17, CXCL2, TNFα, IL-6* accordingly, *n* = 5 mice. **g** *IL-1β* excretion was monitored by ELISA. *a p*-value of the infected-untreated group in comparison to the phage-treated group <0.001, *n* = 5 mice. Data are presented as mean ± standard deviation (SD). Student's *t* test two-tailed unpaired was used, \**p*-value < 0.05, \*\**p*-value < 0.001, and ns stands for "not significant". IL-1β, interleukin-1β. Source data are provided as a Source Data file.

## Table 4 | Mice groups in the in-vitro immune response experiment

| Time point[a] | Group name | Number of mice | Bacterial injection | Phage treatment |
|---|---|---|---|---|
| Day 3 | Sham-injected control | 3 | No | No |
| Day 3 | Infected-untreated | 3 | Yes | No |
| Day 5 | Sham-injected control | 3 | No | No |
| Day 5 | Phage-control | 3 | No | Yes |
| Day 5 | Infected-untreated | 3 | Yes | No |
| Day 5 | treated group | 3 | Yes | Yes |
| Day 7 | Sham-injected control | 3 | No | No |
| Day 7 | Phage-control | 3 | No | Yes |
| Day 7 | Infected-untreated | 5 | Yes | No |
| Day 7 | treated group | 5 | Yes | Yes |

The groups of mice were used to assess the immune response during the anti- *Cutibacterium acnes* phage therapy. After treatment, each group was euthanized at the "Time point," and the following immunological parameters were evaluated: neutrophil and macrophage infiltration, cytokine and chemokines gene expression, and IL-1β secretion.
aThe day the mice were ethnized and assessed.

engineering[41]. Moreover, lytic phages exist against most bacteria, indicating that phages could overcome any defense system, and the arms race between bacteria and phages is constant.

In the current study, we failed to detect any *C. acnes* strain resistant to antibiotic and phage therapy. However, the existence of such mutants should not be excluded, and the search for additional phages capable of providing extensive coverage should remain ongoing. Moreover, phage and antibiotic strains can be isolated in the lab, followed by phage isolation or "training" against them in an iterative manner, which will significantly expand the phage banks and collections.

The eight phages isolated herein are from the *Siphoviridae* family (Fig. 1). Interestingly, to the best of our knowledge, the numerous anti-*C. acnes* phages described to date belong to this family, with slight genomic variance[21,22,33–36,42]. These observations raise unanswered questions regarding the existence of *C. acnes* phages from other families; why do they remain rare even if they do exist? Furthermore, what makes the anti-*C. acnes Siphoviridae* so dominant in the environment?

Moreover, the narrow range of phages seems unique to *C. acnes*. For instance, the phages of *Propionibacterium freudenreichii*, a close bacteria to *C. acnes* and an integral part of cheese manufacturing, while not found in humans[43], are more diverse than the *C. acnes* phages[44]. One possible explanation for this observation is that the "small niche" hypothesis is based on the fact that the most common niche, *C. acnes*, is found in the pilosebaceous unit, possibly hindering the penetration of other types of phages[36]. However, given that we isolated anti-*C. acnes* phages from the saliva and the numerous oral-microbiome studies showing the presence of *C. acnes* in other niches contradict this hypothesis. Another possible explanation is the "bottleneck" hypothesis, which claims that evolutionary limitations have allowed the survival of only one family of phages. Further research using *Siphoviridae*-resistant mutants to select other phages is required to clarify this phenomenon.

Previous studies have assessed the activity and efficacy of injected anti- *C. acnes* phages in mouse models[17–19]. However, to the best of our knowledge, only injectable phages have been previously examined, and the current work is the first report on the topical application of such phages, which is considerably more relevant to clinical settings[9]. Moreover, we found that phages are relatively stable in Carbopol gel (Supplemental Fig S4). Furthermore, in agreement with previous results with anti-*E. coli* phages[45], we demonstrate here that phages in Carbapol gel can penetrate mice skin after a clinically relevant time of 6 h (Supplemental Fig S5). Moreover, this indicates that in the in-vivo model, the observed phage penetration was independent of the needle tract caused by the injection on the first day (Fig 3a–b), further supporting the possibility of topical phage application in humans.

Establishing an acne model using mice could pose a substantial challenge, given that acne is solely a human disease[46]. Herein, we successfully induced *C. acnes*-derived skin lesions by administering two consecutive injections of *C. acnes* strain #27, isolated from a patient with severe acne vulgaris, with artificial sebum applied daily. The lesions were not established when injecting saline instead of bacteria, further validating this model. The lesions did not exhibit precise clinical and histopathological characteristics of human acne vulgaris. While in acne vulgaris comedones are typically seen, the lesions in our model had a deep subcutaneous infiltrate, probably owing to the intradermal injection of *C. acnes*, resembling inflammatory nodules seen in severe acne vulgaris[1,2]. However, our objective in establishing the model was to assess phage therapy, which reduced the bacterial levels and, consequently, the severity of the lesions. Despite minor differences, we hypothesize the occurrence of a similar process in human acne vulgaris lesions. Another limitation of the model was the tendency of mice to self-recover after a few weeks, limiting the experiment to a relatively short period of five days compared to the weeks and months of acne vulgaris treatments in humans. Nevertheless, despite this short treatment period, the treated group

exhibited significant improvement in lesions considering both bacterial load and inflammatory parameters compared with the untreated group. Neutrophil migration to the lesion was reduced, as well as *IL-6, IL-17, CXCL2, TNFα* gene expression in the phage treated group in comparison to the infected-untreated group.

These specific cytokines were chosen due to their involvement in pro-inflammatory processes, as well as macrophage and neutrophil function[47–51].

*IL-6* is a pro-inflammatory cytokine secreted by non-hematopoietic cells as well as by different immune cells[47] playing a role in the differentiation of T-helper 17 cells (Th17)[47]. Another cytokine we assessed, *IL-17* is secreted by Th17 and γδ T-cells[49,52], a subset of T-cells important for the initiation of the inflammatory response[52]. This cytokine mediates differentiation and activation of macrophages[48], as well as recruitment and activation of neutrophils in the tissue[49]. Part of its effect is facilitated through the induction of *CXC* chemokines[49], such as *CXCL2* from non-hematopoietic cells. *CXCL2* attracts neutrophils to the tissue[50] and enables their passage through epithelial tight junctions[50]. Finally, *TNFα*, produced by activated macrophages, is responsible for a diverse range of cell signaling cascades, leading to necrosis or apoptosis[51]. These inflammatory processes contribute to the pathogenesis of acne vulgaris[53]. Moreover, the phage treatment seems safe, as no induction of inflammatory response was noted by applying phages to healthy skin (Figs. 5–6).

Strain #27, used to induce the lesions in the present model, was not antibiotic-resistant. However, given that the main objective of the present study was to prove the concept of phage treatment, we did not test a combined antibiotic-phage therapy, which could be superior to phage therapy alone. In addition, the in-vitro analysis revealed no correlation between the phage and antibiotic sensitivity of the *C. acnes* strains (Table 3 and Supplemental Table S3). Therefore, we believe that similar results could be observed using an antibiotic-resistant strain found to exhibit in-vitro sensitivity to phages.

Taken together, our results support the notion that phage therapy has the potential to offer an additional, efficient, and safe treatment strategy for acne vulgaris and suggest that further clinical trials need to be undertaken to validate these findings, and determine the patient population that would benefit from phage therapy.

# Methods
## Study approval
Clinical and demographic information collection for this study was approved by the local ethics review board, Hadassah Medical Center Helsinki Committee (HMO-0073-19). Clinically isolated bacteria and phage sample collection for this study were approved by the local ethics review board, Hadassah Medical Center Helsinki Committee (HMO-0212-17) Informed consent was given by patients and their parents in case of a minor participant.

## Sampling from patients
The *C. acnes* strains used here and previously described[11], as well as phages, were sampled from acne vulgaris patients > 10 years of age at the Dermatology Clinic at Hadassah-Hebrew University Medical Center as approved by the local ethics review board (HMO-0073-19). According to the ethics' committee request, data about participants was amended, so that individuals cannot be recognized, and data is presented only in aggregates. Their acne lesions were cleaned with 70% isopropyl alcohol wipes, and comedonal or pustular content was squeezed manually or punctured with a sterile needle. The content was collected on a sterile swab (Copan ESwab®, Murrieta, CA) and inoculated onto the surface of culture plates containing Wilkins Chalgren agar (Oxoid, Hampshire, England), supplemented with furazolidone to inhibit the growth of staphylococci.

## Isolation and handling *C. acnes* strains
The prepared plates were incubated under anaerobic conditions for 48 h to 1 week and examined daily for the appearance of typical *C. acnes* colonies. Then, isolation streaks were performed to obtain single colonies sampled to a new plate and for microscopy visualization. To confirm that the observed colonies belonged to *C. acnes*, a matrix-assisted laser desorption ionization-time of flight (MALDI-TOF) analysis was performed[54]. Strains that MALDI-TOF did not recognize were examined by a 16S PCR analysis using Ilumina's universal primers, followed by Sanger sequencing[55]. In addition, an SLST analysis was performed as described[31] using the forward primer: 5'-CAGCGGC GCTGCTAAGAACTT-3' and the reverse primer: 5'-CCGGCTGGCAAATG AGGCAT-3'. The PCR product was sequenced at the Hebrew University sequencing unit and compared with known SLST types (http://medbac.dk/ SLST).

The bacterial strains were grown throughout the experiment in Wilkins Chalgren broth (Difco, Sparks, MD) at 37 °C under anaerobic conditions and stored at −80 °C in glycerol (25%) until use. Bacterial concentrations were evaluated using 10 μl of 10-fold serial dilutions plated on Wilkins agar plates under anaerobic conditions. Colonies were counted after 48 h at 37 °C, and the number of CFU/ml was calculated.

To assess bacterial growth kinetics curves in the presence or absence of phages and antibiotics, logarithmic ($10^7$ CFU/ml) *C. acnes* cultures were incubated in a 200 μl of Wilkins growth media in a 96 wells plate. The growth kinetics of the cultures were recorded using a plate reader (Thermo Fisher, Waltham, MA) at 37 °C in anaerobic conditions every 20 min for 24–48 h.

## Antibiotics
Susceptibility tests were performed as described previously;[11] resistance to clindamycin was defined at a minimum inhibitory concentration (MIC) > 2 μg/ml, tetracycline at a MIC > 4 μg/ml, erythromycin at a MIC of ≥0.5 μg/ml, and doxycycline and minocycline at a MIC ≥ 1 μg/ml resistance was defined according to definitions used in previous studies, and the CLSI guidelines)[11,56,57]. In this study, antibiotic concentration levels were set to the relevant breakpoints for each antibiotic agent.

## Phage isolation and characterization
Phages were isolated from the same skin swabs described above bacterial isolation and from saliva samples – collected from the same patients using a sterile swab and then diluted in 1 ml 0.9% NaCl. The standard double-layered agar method was used[58]. Briefly, samples were dissolved and mixed in 5 ml of phage buffer (150 mM NaCl, 40 mM Tris-Cl [pH 7.4], 10 mM MgSO4), followed by centrifugation (centrifuge 5430 R, rotor FA-45-24-11HS; Eppendorf, Hamburg, Germany) at 10,000 × g for 10 min. The supernatant was filtered twice through a 0.45-μm and then 0.22-μm pore size (Merck Millipore, Burlington, MA). The filtrate was mixed with exponentially grown bacterial cultures for 24 h at 37 °C in anaerobic conditions. The cultures were filtered and serial diluted, with each diluent incubated in 5 ml soft Wilkins agar (0.6%) containing 0.5 ml of overnight-grown *C. acnes*, plated on an agar plate. Clear plaques were transferred into a broth tube using a sterile Pasteur pipette. The phage stocks were stored in Wilkins at 4 °C.

A PFU count of quantifying the titer of phages and the genome of phages was sequenced as previously described[59]. Briefly, DNA was extracted using a phage DNA isolation kit (Norgen Biotek, Thorold, Canada)[60], and libraries were prepared using the Illumina Nextera XT DNA kit (Illumina, San Diego, CA). Normalization, pooling, and tagging were performed with a flow cell (1 × 150 bp single-end reads). Sequencing was performed using Illumina NextSeq 500 platform at the Hebrew University of Jerusalem sequencing unit at the Hadassah Campus. Trimming, quality control, read assembly, and analyses were

performed using Geneious Prime 2021.2.2 and its plugins (https://www.geneious.com). Assembly was performed using the SPAdes plugin of Geneious Prime. Annotation was performed using RAST version 2 (https://rast.nmpdr.org/rast.cgi) and the Phage Search Tool Enhanced Release (PHASTER) (HTTPS://phaster.ca, accessed on 1 March 2021), and tRNAs were predicted using http://trna.ucsc.edu/cgi-bin/tRNAscan-SE2). The phages were scanned for resistant genes and virulence factors using Abricate (Seemann T, Abricate, https://github.com/tseemann/abricate) based on its databases: NCBI, CARD, Resfinder, ARG-ANNOT, EcOH, MEGARES, PlasmidFinder, Ecoli_VF, and VFDB. All of the isolated phage sequences were uploaded to the NCBI Genebank, association numbers are supplied (Table 2). A phylogenetic tree was constructed using the tree builder function of Genious Prime (Version 2022.2.1, New Zeland) with Tamura-Nei Genetic Distance Model, Neighbor-joining tree method with Bacillus phage E as an outgroup (marked in red).

TEM visualization of phages was performed as described in the microscopy department, the intradepartmental unit of Hebrew University[59]. Briefly, a sample of 1 ml with $10^8\frac{PFU}{mL}$ was centrifuged at 20,000 × g (centrifuge 5430 R, rotor FA-45-24-11HS; Eppendorf, Hamburg, Germany) for 2 h at 18–25 °C. The supernatant was discarded, and the pellet was resuspended in 200 µl of 5 mM MgSO₄, spotted on a carbon-coated copper grid with an addition of 2% uranyl acetate, and incubated for 1 min. The excess was removed, the sample was visualized using a TEM 1400 plus Joel, Tokyo, Japan), and a charge-coupled device camera (Gatan Orius 600) was used to capture images.

### Phage stability assessment

The eight phages' activity was assessed following incubation in Carbopol gel for 30 days, with samples taken on days 1, 5, 10, 15, 20, and 30. Each day, PFU was calculated from a gel tube and a matching control tube.

### Phage therapy of C. acnes-induced lesions of mice

A virulent C. acnes isolate (strain #27)[11] was arbitrarily selected for this model after screening of several bacterial strains. The bacteria was grown for three days to reach a concentration of $10^9\frac{CFU}{mL}$. Briefly, bacteria (50 µl) were intradermally injected at two intervals of 24 h into the right and left sides of the back of 34 8-week-old, albino ICR strain, female mice, as described[24]. As a negative control, four mice were sham-injected with saline. Female mice were used based on a previous publication describing this model[24], mice were housed at 21–24 °C, 30–70% humidity, with light in the room for 12 h, between 7:00–19:00, and dark for 12 hours from 19:00–7:00. Then, 20 µl of artificial sebum (17% fatty acid, oleic acid, triolein, 25% jojoba oil, and 13% squalene) was applied immediately after the first intradermal injection and reapplied daily for the duration of the experiment, as described previously[24]. The mice were euthanized and cleaned with chlorhexidine three days post-last gel administration. The sample size was calculated with the size-calculator ClinCalc (https://clincalc.com/stats/samplesize.aspx), with a preliminary estimation of a score of 1.9 ± 0.4 AU in the control group and 1.5 AU in the treated group (n = 16).

At day 3, infected mice were randomly assigned into two groups of 17 mice, with each group treated topically for five days with either 0.5 ml of 2.5% Carbopol with or without phage FD3 at concentration of $10^9\frac{PFU}{ml}$. The gel was not removed during the following applications. Different groups were separately housed to prevent the transfer of phages between them.

In another experiment, 34 mice in 10 different groups of sham-injected, phage control, infected-untreated, and phage-treated mice were euthanized at various time points, as described (Table 4). Sample size was determined according to a peliminary estimation was 45 ± 14 %Noutophil in the infected-untreated group and 21% Noutophil in the treated group (n = 5, ClinCalc sample size calculator), in order to gain other time points, a minimal n = 3 was used

sham-injected and phage control mice were injected with saline for two consecutive days, and sebum was applied daily. From day 3, only the phage control was treated with 0.5 ml of 2.5% Carbopol gel with phage FD3 at a concentration of $10^9\frac{PFU}{ml}$. Infected mice were infected as described above, randomly allocated to Carbapol gel with or without FD3 phage, and treated as described above.

### Analysis of C. acne-induced lesions

All lesions were photographed daily, and the degree of clinical status was scored (Supplemental Fig. S6) using three clinical parameters: (1) Lesion diameter was measured using an electronic caliper (Winstar), (2) Degree of elevation/papulation of lesions, and (3) Presence and severity of eschar within lesions.

Histopathological evaluation was performed on 3 mm punch skin biopsies from the inflammatory lesions at the end of the treatment period. Several representative biopsies were used for histopathological evaluation, and other tissue samples were homogenized in sterile phosphate-buffered saline (PBS) using stainless steel beads and a bullet blender tissue homogenizer (Next Advance) for bacterial and phage assays. The presence of bacteria at the lesion's site was validated by PCR using the SLST primers: forward primer 5′-CAGCGGCGCTGC TAAGAACTT-3′ and reverse primer 5′-CCGGCTGGCAAATGAGGCAT-3′. The presence of FD3 phage was assessed using forward primers 5′-TG ATGCTGTAGGTGGCTGTG-3′ and reverse primer 5′-CCGAGACGAAAT GACCACCA-3′, designed using Primer-Blast (https://www.ncbi.nlm.nih.gov/tools/primer-blast). In addition, PFU and CFU counts were determined to quantify viable FD3 and C. acnes #27, respectively, in lesions.

In addition, pathological skin biopsies were soaked in formalin, then trimmed and embedded in paraffin. Following these steps, the paraffin slide was sectioned and stained with hematoxylin and eosin. Evaluation of lesions was performed at the Histology Laboratory of the Animal Facilities of the Faculty of Medicine of Hebrew University. Based on a pathologist report, histopathological images were scored as mentioned above, and the lesion area was calculated using ImageJ software (V.1.53).

Neutrophils were quantified in the lesion by immunofluorescence staining. The slides were deparaffinized with xylene, and 100%, 95%, 80%, and 70% ethanol before washing three times with PBS. Slides were then blocked in blocking buffer (PBS, 10%FCS, 10% BSA, 2% Triton X-100) for 1.5 h at 18–25 °C and incubated with primary rat anti-mouse Ly-6G clone 1A8, diluted 1:100 (BD Biosciences, Franklin Lakes, NJ) overnight at 4 °C. Following three washing steps in PBS, the samples were incubated with a secondary antibody goat anti-rat IgG (Invitrogen, Waltham, MA), diluted 1:200 in blocking buffer for 1 h at 18–25 °C, washed three times, stained with DAPI, and mounted. As a negative staining control, the primary antibody was omitted and replaced by a blocking buffer. Signals were visualized, and digital images were obtained using a confocal laser microscope (Nikon Yokogaha W1 spinning disk, Tokyo, Japan).

In addition, neutrophils and macrophages were quantified by flow cytometry. Extraction of immune cells from skin biopsies was performed using the protocol described by Lou et al.[61], with some modifications to the Dispase II concentration and incubation time. Briefly, 1 × 1 cm of skin was sampled from the acne-induced lesions after mice euthanization. The skin was washed in Hanks' Balanced Salt Solution (HBSS) three times, cut diagonally into four pieces, and incubated with 8 mg/ml Dispase II (Merck, Kenilworth, NJ) for 12 h. Following dermis and epidermis separation, the dermis was cut and placed in 3.5 ml of Dermis Dissociation Buffer composed of 100 µg/ml of DNASE I (Merck, Kenilworth, NJ) and 1 mg/ml collagenase P (Merck, Kenilworth, NJ) in Dulbecco's Modified Eagle Medium (DMEM/high glucose) (Merck, Kenilworth, NJ) for 1 h. Suspensions were passed through a 40-µm strainer into a 50 ml tube and rinsed again in 12 ml of DMEM with 10% fetal bovine serum (FBS) (Merck, Kenilworth, NJ) in 15 ml tubes, centrifuged at 400 × g for 5 min at 4°C, supplied with 2 ml of staining

buffer[61] composed of PBS with 2% fetal calf serum (Merck, Kenilworth, NJ), and fixated with 250 µl BD Cytofix™ (BD Biosciences, Franklin Lakes, NJ).

Single-cell suspensions were incubated and labeled with the following antibodies obtained from BioLegend (San Diego, CA) at a 1:100 dilution: CD115 (AFS98), CD45 (30-f11), CD64 (X54-5/7.1), CD11b (M1/70), LY6C (HK1.4), LY6G (1A8), and Zombie UV™ for dead cell exclusion Antibodies used herein are listed in Table S5. Following membrane staining, cells were fixed using a Fixation/Permeabilization Solution Kit (BD) according to the manufacturer's instructions. Flow cytometry was performed using Cytek® Aurora (Cytek, Fremont, CA), and data were analyzed offline using FlowJo 10.7.2 (BD Biosciences, Franklin Lakes, NJ). We described the gating strategy for both cell populations, neutrophils were defined as $CD45^+CD11b^+Ly6G^+$ [62], and macrophages were defined as $CD45^+CD11b^+CD64^+$ [63] (Supplemental Fig. S8B).

Gene expression of cells in lesions was assessed by quantitative PCR. Briefly excised skin fragments were homogenized in 300 ml TRI reagent (Sigma-Aldrich, Burlington, MA) using an electric homogenizer (IKA labortechnik, Staufen, Germany), and RNA was extracted using an RNA purification kit (Qiagen, Hilden, Germany). cDNA synthesis was performed using the qScript cDNA Synthesis Kit (Quanta-BioSciences, Gaithersburg, MD). RT-PCR reactions in 20 µL volumes were performed using Power SYBR Green PCR Master Mix (Quanta-BioSciences, Gaithersburg, MD) with 10 min at 95 °C, 40 cycles of 15 s at 95 °C, and 60 s at 60 °C. The primer sequences are listed in Table S6.

Secreted levels of IL-1β were measured using ELISA kits Invitrogen (Invitrogen, Waltham, MA).

## Assessment of Ex-vivo skin penetration by phages
Phage penetration assessment was done based on the previously described Franz Diffusion Cell method[45] with 0.4µm transwell system (SPLInsert Hanging, SPL Life Sciences, Kyonggi-do, South Korea)[64,65]. Briefly, healthy full-thickness skin fragments were harvested from 8-week-old albino ICR strain female mice and placed into the donor compartment of the transwell stratum corneum facing up. The device was placed in a 24 wells plate with 1 ml Wilkins medium. FD3 phage ($10^9 \frac{PFU}{ml}$) in Wilkins medium or Carbapol gel, mixed with Allura Red dye (15 mg/ml in DDW, Sigma-Aldrich, Burlington, MA), were spotted on the skin (drops of 10 µl). Drops of Wilkins medium mixed with the dye were used as a negative control. The samples were incubated for 24 h at 18–25 °C followed by PFU count from the recipient compartment at various time points. At the endpoint, the optical density of the Allura Red at the recipient compartment was assessed (OD 505 nm[66]). Additional controls were transwells without skin and transwells with a parafilm fragment instead of the skin.

## Statistical analysis
GraphPad Prism 8.0.2 (GraphPad Software, Inc., La Jolla, CA) was used for statistical analysis and graph preparation. Significance was calculated using the Student's t-test, two-tailed unpaired p-values (significance level: $p$-value $< 0.05$). The results were the mean of at least three independent experiments. ± stands for standard deviation. Outliers were defined (but not removed) if their value was more than three SD from the mean.

## Figure assembly
Figures were assembled using Illustrator (Version 24.1, Adobe, San Jose, CA), in Fig. 3 and S5, the illustrations were made using Biorender (biorender.com), publication license was granted (Supplementary file).

## Reporting summary
Further information on research design is available in the Nature Portfolio Reporting Summary linked to this article.

## Data availability
All data presented herein are available in the main text, supplementary materials, source files, or data depositories. The phage sequences generated in this study were deposited in GenBank (https://www.ncbi.nlm.nih.gov) with the accession codes described in Table 2. Images not displayed in the Supplementary section can be found in the BioStudies data repository (https://www.ebi.ac.uk/biostudies/studies/S-BSST889). Source data are provided in this paper.

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

## Acknowledgements

We would like to thank Dr. Simon Yona for advising with flow cytometry, and supplying the materials for the flow analysis, the Core Research

Facility of the Hebrew University, Ein Karem Campus, Dr. Abed Naser-eddin and Dr. Idit Shiff for the deep sequencing, and Dr. Eduard Berenshtein for the TEM. We are grateful to Mariana Scherem for helping us prepare our in-vivo experiments. Funding: United States–Israel Binational Science Foundation Grant #2017123 granted to Prof. Ronen Hazan. Israel Science Foundation IPMP Grant #ISF1349/20 granted to Prof. Ronen Hazan. Rosetrees Trust Grant A2232 was granted to Prof. Ronen Hazan. Milgrom Family Support Program granted to Prof. Ronen Hazan. Gishur Fund of Hadassah Medical Center granted to Dr. Vered Molho-Pessach. George and Linda Hiltzick's donation to Hadassah Medical Center was granted to Dr. Vered Molho-Pessach.

## Author contributions

A.R., S.C.G., V.M.P., and R.H. conceptualized and designed the research. A.R., V.L., L.S., R.L., K.Z., N.J., and M.M. performed the in-vitro experiments. S.S.L. and V.M.P. collected the samples from the patients and V.L. and S.S.L. isolated the bacteria and phages. A.R. and C.R. designed and performed the in-vivo and ex-vivo experiments and analyzed the data. S.A.O. and R.H. did the phage genome analysis. T.S. helped with the statistical analyses. A.R. and R.H. wrote the paper. A.W., S.C.G., V.M.P., and R.H. mentored, instructed, and revised the manuscript.

## Competing interests

The authors declare no competing interests.
