## [Peer Review File · Nature Communications]

REVIEWER COMMENTS

Reviewer #1 (Remarks to the Author):

In this study, the authors isolated new *C. acnes* phages and have found that they are useful in killing *C. acnes* in vitro. By applying the phage-cream formulation to *C. acnes*-infected mice, the inflammatory reaction was significantly alleviated. And this may be due to the reduction of neutrophil infiltration. However, some descriptions of results and conclusions appear too superficial and general, and thereby generate, as far as I can see, wrong statements.

First, a general comment: There are errors in the use of English throughout that require attention and editing. Throughout the manuscript, some re-arrangement of the content and inclusion of some more linking sentences might help to improve cohesiveness.

My comments that follow are more general, but follow the manuscript seriatim, focusing on the results, underlying assumptions, and conclusions, which, to my mind, need more critical interpretation.

Major comments:

1. Line 107-111: please expand the discussion.
2. Line 112-113: The author should provide the annotation data of the open reading frames in the supplementary data to prove their statements.
3. As all these phages were isolated from Israel (I assumed), the authors may expand their discussion by comparing their phage to other phages discovered.
4. Line 118 and throughout the manuscript: Length and diameter: the standard error should be added. Which virus family do all these phages belong to (according to TEM results)?
5. Line 121-122: (8) ??
6. Line 126: No data were showing that the bacteria were inhibited by "all phages". All of the individual data should be added in Supp Fig.
7. Line 127-130 and Figure 2D: How come the control group dropped compared to the CM group?
8. Line 138-139: Fig 2C did not show the results of what the author stated: "Moreover, the four phage-resistant strains were susceptible to all five tested antibiotics"
9. Figures should be numbered in the order they are first mentioned in the text (i.e. 2D and E were cited before 2C in the manuscript).

10. Line 133-136; this section regarding the SLST type can be moved to legends of Supp Fig.
11. Line 138-139: "Moreover, the four phage-resistant strains were susceptible to all five tested antibiotics" The author can expand the discussion on this phenomenon. Does that mean the phage is only useful to antibiotic-resistant bacteria but not the sensitive ones? Phage-resistant bacteria? Perhaps this is related to the patient's status where the bacteria isolated?
12. Line 144: Bacteria strain #27 are antibiotics (Abx)-sensitive bacteria (according to Supp Table 2). Therefore routine Abx treatment is already useful to alleviate the pathology. Phage therapy is raising interest because of the "Abx-resistant bacteria". The authors should also test the phage cream on mice with Abx-resistant bacteria infection.
13. Line 145: How did the author determine "high in vitro efficacy"??? as the author only state "S" or "R" in the results.
14. Line 146: "with no difference over a cocktail of phages" How was this conclusion drawn from Supp Table 2?
15. Line 147-148: Data? Did the author also formulate the rest of the phages?
16. Line 147-148: Also, when formulating a cream-based product, especially containing a biological-active component, we usually consider their stability as well as microbial content upon storage. (different cream base may affect the viability of the phage) Although this is not the aim of this study, in my opinion, the author may consider discussing it.
17. Line 165 and throughout the manuscript: $1.4 \cdot 10^4$ should be 1.4×10^4 .
18. Line 162, 169-171: "(for more details see "Assessment of Phage Clinical Efficacy" section in Materials and Methods and Fig. 7)" this statement should not be presented in the result section.
19. Fig 4: Statistics should also compare between days (e.g. before the treatment) in the same group.
20. Try to simplify Fig 3.
21. Fig 3 and Fig 4 should be combined into one figure
22. Line 178: standard error?
23. Figure 4B-G: Please add standard errors and units in the text box
24. Figure 4H-K: Why only days 4 and 8 were shown? Is there any reason for using the combined score (Figure 4K)?
25. Line 196-197: what about the comparison to the beginning of the experiment (before the phage treatment?)
26. Fig 204-208: Figures should be numbered in the order they are first mentioned in the text
27. Line 209: "histological score based on the pathological report" what does this mean?
28. Line 210: average area "of the lesion"... sounds better; Also, please add the standard error.

29. Figure 5 and Line 203-204: "...neutrophils surrounded by macrophages and rare lymphocytes,..." None of these cells can be seen in the figure (in such a low power field). The author can only suggest cellular infiltration from the image presented in Fig 5. The author should provide images at a higher power field.

30. Figure 5.:Any reason for combining the score together? How come the sample size differs in the combined score from others?

31. Figure 5: Neutrophil staining has to be done to confirm that the infiltrated neutrophils are indeed resolved by the treatment. (More evidence is needed to support the author's theory!)

32. Line 219: "Three mice were sacrificed during the experiment" Are these mice from those used in Figure 5 or are these extra mice?

33. Figure 6E: What does this figure imply?

34. Table 2: What is the phage used here?

35. Line 265-266; 268-269; 270-275: Because the bacteria were sensitive to either phage or Abx, so this is foreseeable that the bacteria is sensitive to combinational therapy. Therefore, the conclusion drawn by the authors may not be correct (also in the abstract).

36. Line 265-266: In vivo or in vitro model using *C. acnes* strain resistant to both phages and Abx should be used to confirm this statement.

37. Line 309: "Table XX"???

38. Figure 6: A phage control (uninfected mice using the cream) should also be investigated in the study to exclude the baseline level of immune activation by the phage

39. At last, the author should also expand their discussion on phage resistance, as this is another problem when using phage therapy. (In this study there are also phage-resistant bacteria)

40. The discussion would flow more smoothly if it were more integrated. The current discussion section sounds more like a result section to me.

41. Was an ELISA ever done to look at serum and cell-released levels of IL-1B? This may provide information about the general state of the mice.

Material and methods

42. Line 381: skin sample or skin swab?

43. Line 432: 5 mM

44. Regarding sample collection and phage isolation: Was IRB approval obtained to collect the sample from patients? How was the sample collected? What are the characteristics of these patients? Are they Abx responsive? Did the saliva also collect from *acnes* pts? And more details are needed.

45. When were the uninfected mice euthanized? How were they treated?
46. Line 461: What liquid media? How was the skin incubated?
47. Please move Figure 7 to Supp Fig.
48. Line 486-487: How was the phage primer designed?
49. Line 520: "RT-PCR For RNA isolation": RT-PCR was not used for RNA isolation
50. Line 522: Where was Pearson analysis used?
51. How was the sample size determined?
52. Line 526-527: cycling threshold is not "DCT" (should be "Ct")
53. Line 526-527: please spell out "2-DDC" and use the correct abbreviation.
54. Line 534: Was this the approval for the animal experiment?
55. Table S4: Were these primers newly designed for the study? If so, please provide more details for each of them.
56. Line 450-453: But there are five samples on day 5 (Fig 6D). Also, where are the data from day 7?

Figures and Figure Legends:

57. Table 2: which phage?
58. Table 2 legend: "Some isolates were verified as non *C. acnes* and therefore some numbers are missing" what does this mean?
59. Figure 1: The figure is difficult to see.
60. All figures: Low-resolution images are not acceptable.
61. Figures 2C and F are not necessary; or consider moving them to the Supp Fig.
62. Figure 3: Did image capturing and pathology study also take place on day 8? (as shown in Supp Table 3)
63. Line 805-806: Again, the macrophage and neutrophils cannot be distinguished from the figure.
64. Figure 6A-C: The group name should be added to the figure.
65. Figure 6A-C: Gating strategies should be added in Supp Figure
66. Figure 6F: I can't see from the figure which group was used as a normalizer to calculate relative expression? (The value of the normalizer seems to differ in each group?)
67. More details are needed in all the figure legends (both main Figure and Supp Figure)

68. Supp Table 3: The author should take all the pictures from the same angle, light source, background, etc. For example, images on day 8 from control animals #15 and #16 are totally different than the rest of the picture, which made us difficult to interpret the data. Other examples such as Day 7 from treated animal #1; Day 8 from treated animal #16; as well as others!

69. Supp Table 3: Also, the author should provide a scale bar (e.g. ruler with a scale bar next to the lesion site) when taking the picture.

70. Supp Fig 5: How were the macrophages gated?

71. Supp Fig 5: It is interesting that there is no significant difference between days 3 and 5 (in the FD3 group) when the level is apparently high on day 5. Was this comparison done?

72. Supp Fig 5 legend and Supp Fig 5: In the legend the author stated "monocytes", however in the figure it showed "macrophages"

73. All the tables and figures need to be reproduced to provide clear contents (as well as a clear and informative figure legend)

Other minor points for improvement:

74. An abstract should be typed as a single paragraph.

75. The author please recheck the manuscript carefully for spelling mistakes. For example:

Line 341: medhanisms or mechanisms? Crrrossig or crossing?

Line 427: Phylogeneteic or phylogenetic?

Line 299: type in italics

76. Line 242: Figure 1F should be Figure 6F.

Reviewer #2 (Remarks to the Author):

This work shows that use of phages in a mouse model of Cutibacterium acne brings a benefit, although not a notably large benefit.

The work is well designed and executed, although its subsequent write-up is less attractive.

The manuscript needs to be carefully and thoroughly edited to achieve correct scientific English. There are numerous mistakes in spelling, punctuation, grammar, and incorrect use of capital letters. The sentence in line 450 no verb.

The term 'sacrifice' which generally means 'slaughtering an animal or person as an offering to a deity' is an unacceptable euphemism for "kill"; "sacrifice" should be replaced throughout with a less euphemistic term such as kill or euthanase.

The main issue with this work seems to be around the animal model. The paper would be improved by:

(i) Discussion justifying the suitability of the model, and the route of delivery of the bacteria. Why is it injected, and not, for example, applied to scarified skin? Given that the model required two injections of a relatively large volume of bacteria at high concentration, is the model really representative of what happens in acne?

(ii) Improved description of how the phage was applied. Neither Materials and Methods nor Figure 3 gives enough detail about how the phage was applied. Was the gel simply smeared over the skin lesion? If so, what volume of gel? Was any intervention used to keep the gel on the lesion and stop the mice from rubbing it off? If not, for how long was the gel in place on each lesion?

Dear Dr. Hayleah Pickford, Editor of Nature Communications,

Please find our point-to-point answers to the reviewers' comments on our manuscript, termed "*Toward phage therapy for acne vulgaris: Topical application in a mouse model of Cutibacterium acnes-induced acne-like lesions*" NCOMMS-22-20 by Rimon *et al.* We want to thank both the reviewers and you for the constructive comments that improved our manuscript significantly. Accordingly, as requested, we revised the manuscript and added experiments supporting our conclusions, including immunofluorescence and ELISA. In addition, we sent the manuscript again for language editing (see attached certificate) and hope that the language is now much better.

REVIEWER

COMMENTS

Reviewer #1 (Remarks to the Author): In this study, the authors isolated new *C. acnes* phages and have found that they are useful in killing *C. acnes* in vitro. By applying the phage-cream formulation to *C. acnes*-infected mice, the inflammatory reaction was significantly alleviated. And this may be due to the reduction of neutrophil infiltration. However, some descriptions of results and conclusions appear too superficial and general, and thereby generate, as far as I can see, wrong statements.

➤ We thank the reviewer for these comments. Accordingly, we elaborated the results thoroughly and altered the general conclusions to state our findings accurately (see details below).

First, a general comment: There are errors in the use of English throughout that require attention and editing. Throughout the manuscript, some rearrangement of the content and inclusion of some more linking sentences might help to improve cohesiveness.

➤ We apologize for these oversights. Given that we are non-native English speakers, we sent the manuscript for professional editing. However, the editing was unsatisfactory. Hence, we sent the current version of the manuscript for English editing to Wiley Editing Services (<https://wileyeditingservices.com/en/>), see attached certificate). We truly hope that the revised manuscript meets the journal's standards.

My comments that follow are more general, but follow the manuscript seriatim, focusing on the results, underlying assumptions, and conclusions, which, to my mind, need more critical interpretation.

Major

comments:

1. Line 107-111: please expand the discussion

➤ We expanded the discussion in terms of differences between the anti-*Cutibacterium acnes* phages in lines 113-125 and Fig. 1B-C.

2. Line 112-113: The author should provide the annotation data of the open reading frames in the supplementary data to prove their statements.

➤ Based on the reviewer's comment, we provided annotation data in Supplemental Table 2

3. As all these phages were isolated from Israel (I assumed), the authors may expand their discussion by comparing their phage to other phages discovered.

➤ We thank the author for raising this interesting point. Accordingly, this has been elaborated in lines 390-405

4. Line 118 and throughout the manuscript: Length and diameter: the standard error should be added. Which virus family do all these phages belong to (according to TEM results)?

➤ We added standard deviations throughout the manuscript. We added Supplemental Fig. S1 with an example of transmission electron microscopy (TEM) images for all 8 phages, along with the measured length and capsid diameter for each phage in Table 2. Given the high genetic similarity, we did not observe a significant difference in TEM. According to sequence and TEM findings, isolated bacteriophages belong to the Siphoviridae family.

5. Line 121-122: (8)

➤ This was a typographical error, which has been deleted in the revised manuscript.

6. Line 126: No data were showing that the bacteria were inhibited by "all phages". All of the individual data should be added in Supp Fig.

➤ We are unsure if we comprehensively understand this comment. We did include the results in Table S2 in our manuscript, referred to in line 126. Nevertheless, in the revised manuscript, we included plaque images of each phage for each bacteria (Fig S2A), indicating that phage-resistant strains are indeed sensitive to clindamycin (Fig S2 B-D), along with the kinetics of each phage toward selected bacteria (Fig S2 F-M).

7. Line 127-130 and Figure 2D: How come the control group dropped compared to the CM group?

➤ Currently, we are unable to explain this finding; however, we can confirm that this phenomenon is reproducible within this strain, in which clindamycin was shown to inhibit autolysis^{1,2}. It should be noted that this is not a general phenotype but a strain-specific effect. It needs to be determined whether clindamycin causes direct or indirect inhibition of autolysin, an enzyme known to be involved in autolysis.

We elaborate on this point in the discussion in lines 355-359 in the revised manuscript.

8. Line 138-139: Fig 2C did not show the results of what the author stated: "Moreover, the four phage-resistant strains were susceptible to all five tested antibiotics"

➤ As stated in our answer to comment 6, we added these results in Fig 2D (previously 2C) and Supplemental Figure S2B-D, presenting the susceptibility of phage-resistant *C. acnes* strains to clindamycin and a combination of clindamycin and phage FD3.

9. Figures should be numbered in the order they are first mentioned in the text (i.e. 2D and E were cited before 2C in the manuscript).

➤ Based on the reviewer's comment, we corrected the order of the figure numbers throughout the manuscript.

10. Line 133-136; this section regarding the SLST type can be moved to legends of Supp Fig.

➤ We moved this section to the legend of Supp. Fig S3

11. Line 138-139: "Moreover, the four phage-resistant strains were susceptible to all five tested antibiotics" The author can expand the discussion on this phenomenon. Does that mean the phage is only useful to antibiotic-resistant bacteria but not the sensitive ones? Phage-resistant bacteria? Perhaps this is related to the patient's status where the bacteria isolated?

➤ We detected no clear correlation between antibiotic resistance/sensitivity to phage resistance/sensitivity (Supp Fig S2E). In our collection of 36 strains, we found 21 strains sensitive to both and 4 and 11 sensitive only to antibiotics or phage therapy, respectively. Indeed, we did not detect any strain resistant to both, but we cannot exclude the possibility of such mutant existence that it was not found, perhaps, due to the small size of the present collection. Regarding the question posed by the reviewer, in our opinion, the decision to utilize phages against antibiotic-sensitive bacteria or only in cases of antibiotic resistance should be at the treating clinician's discretion. We elaborate on this point in the discussion in lines 385-389, 439-442 in the revised manuscript.

12. Line 144: Bacteria strain #27 are antibiotics (Abx)-sensitive bacteria (according to Supp Table 2). Therefore routine Abx treatment is already useful to alleviate the pathology. Phage therapy is raising interest because of the "Abx-resistant bacteria". The authors should also test the phage cream on mice with Abx-resistant bacteria infection.

- In the present study, we examined the efficacy of topical phage therapy *in vivo*. Establishing *C. acnes*-induced lesions in mice can be challenging. Mice naturally resist this bacterium and are spontaneously cured following lesion induction without treatment.

Furthermore, our antibiotic-resistant strains were less virulent than sensitive ones for yet unknown reasons. Eventually, after screening several isolated strains and identifying a more virulent strain, and with the addition of human sebum, we finally succeeded in inducing lesions that lasted a few weeks. This virulent strain was antibiotic-sensitive. Given that the objective of the present study was to demonstrate the concept of phage therapy, we did not insist on finding a virulent antibiotic-resistant bacteria. Moreover, based on our findings and those reported previously, there appears to be no clear correlation between antibiotic resistance and phage susceptibility^{3,4}. As described in lines 414-429 in the revised manuscript.

This point was further elaborated in the discussion section of the revised manuscript. Nevertheless, if the reviewers find this assay essential for establishing the validity of the manuscript, we will conduct this experiment; however, considerable time will be needed to identify the best antibiotic-resistant strain for lesion establishment, as described above.

13. Line 145: How did the author determine "high *in vitro* efficacy"??? as the author only state "S" or "R" in the results

- We apologize for the lack of clarity regarding the same. The definitions of S and R were determined by the plaque assay, as described in the Methods section and presented in Supp Fig S2A. The phrase "high *in vitro* efficacy" was clarified in lines 169-171 in the revised manuscript. It is based on the kinetics experiments presented in Fig Supp Fig 2H, demonstrating no bacterial regrow during the 96 h of the assay.

14. Line 146: "with no difference over a cocktail of " How was this conclusion drawn from Supp Table 2?

- Here, perhaps Supplemental Fig.2 was misread as Supp Table 2? In Supp Fig S4B (formerly 2C), one can observe that the phage cocktail lacked apparent advantages over FD3, as it had already eliminated bacterial activity.

15. Line 147-148: Data? Did the author also formulate the rest of the phages?

- Indeed, we assessed the stability of all phages in Carbopol gel, which primarily impacted our decision to select phage FD3 for animal experiments. Accordingly, we compared each phage in Carbopol gel to the Wilkins media tube at 4°C, and we sampled tubes on days 1, 5, 10, 15, 20, and 30. We noted no significant differences between the control and gel over time, *i.e.*, the Carbopol gel did not reduce phage viability compared with Wilkins media. This data was added as Supplemental Fig S4A, in lines, 160-168 in the results section, lines 410-412 in the discussion, and lines 519-522 in the methods section of the revised manuscript.

16. Line 147-148: Also, when formulating a cream-based product, especially containing a biological-active component, we usually consider their stability as well as microbial content upon storage. (different cream base may affect the viability of the phage) Although this is not the aim of this study, in my opinion, the author may consider discussing it.

➤ As mentioned in the previous comment, this point has been addressed in Fig S4A.

17. Line 165 and throughout the manuscript: $1.4 \cdot 10^4$ should be 1.4×10^4 .

➤ This has been corrected

18. Line 162, 169-171: "(for more details see "Assessment of Phage Clinical Efficacy" section in Materials and Methods and Fig. 7)" this statement should not be presented in the result section.

➤ This statement was deleted.

19. Fig 4: Statistics should also compare between days (e.g. before the treatment) in the same group.

➤ In the revised manuscript, statistical analysis between days was incorporated in Fig. 3 (Formerly Fig. 4).

20. Try to simplify Fig 3.

➤ We reduced the redundant wording and elements. Additionally, we rearranged and combined Fig. 3 with Fig. 4 (now numbered Fig. 3). We believe these changes will adequately simplify Fig 3.

21. Fig 3 and Fig 4 should be combined into one figure

➤ We combined the two figures.

22. Line 178: standard error?

➤ Standard deviation (SD) was added throughout the manuscript.

23. Figure 4B-G: Please add standard errors and units in the text box

➤ Units were added to the revised manuscript. Data presented in this figure represent a single score at a specific time for a specific lesion; thus, standard errors could not be added.

24. Figure 4H-K: Why only days 4 and 8 were shown?

➤ We added all time points to Fig. 3 in the revised manuscript.

Is there any reason for using the combined score (Figure 4K)?

➤ To grade the lesion, we used multidimensional parameters, and the score is an integration of these parameters. We believe that a single parameter cannot accurately describe the lesions.

25. Line 196-197: what about the comparison to the beginning of the experiment (before the phage treatment?)

➤ A comparison to the beginning of the experiment was added to Fig. 3 I-L and in lines 261-263 of the revised manuscript.

26. Fig 204-208: Figures should be numbered in the order they are first mentioned in the text

➤ The order of figure numbering has been corrected in the revised manuscript.

27. Line 209: "histological score based on the pathological report" what does this mean?

➤ The score of the histological analysis was based on the interpretation of an independent expert animal pathologist. We established a quantitative scale based on this interpretation, allowing the comparison between the groups, added to Supplemental Figure S7; see below.

Pathologist report	Score
WNR (Within normal range)	1
The subcutaneous fat and skeletal muscle (panniculus carnosus) have a nodular to diffuse subcutaneous infiltration with neutrophils and macrophages.	2
The subcutaneous fat and skeletal muscle (panniculus carnosus) have a nodular to diffuse subcutaneous infiltration with neutrophils and macrophages. There is central necrosis within the infiltrate.	3

28. Line 210: average area "of the lesion"... sounds better; Also, please add the standard error.

➤ We agree with the reviewer's comment and corrected it accordingly. We also added standard deviation throughout the manuscript.

29. Figure 5 and Line 203-204: "...neutrophils surrounded by macrophages and rare lymphocytes,..." None of these cells can be seen in the figure (in such a low power field). The author can only suggest cellular infiltration from the image presented in Fig

- An independent pathologist performed this interpretation. Nonetheless, we revised the sentence to: "A nodular infiltrate was observed, with an area of necrosis (Fig. 4B, C)" in lines 273-274 of the revised manuscript.

5. The author should provide images at a higher power field.

- Fig 4C, F with a higher power field was added.

30. Figure 5: Any reason for combining the score together?

- We believe that score integration adds simplicity and allows clinicians to decide and follow treatments based on the lesion condition. Nevertheless, we also provide each score parameter (Fig 4).

How come the sample size differs in the combined score from others?

- The histological score refers only to mice that underwent histological evaluation, while the clinical score refers to all mice employed in the study. In the revised manuscript, we only presented scores for mice that underwent histological assessment.

31. Figure 5: Neutrophil staining has to be done to confirm that the infiltrated neutrophils are indeed resolved by the treatment. (More evidence is needed to support the author's theory!)

- In accordance with the reviewer's comment, we added immunofluorescence staining for Ly6G+ cells to quantify neutrophils in each group (new figure 5 A-C). The experimental details were added in the results in lines 301-306 and in the methods in lines 559-567 of the revised manuscript. All images can be found in the BioStudies data repository (<https://www.ebi.ac.uk/biostudies/studies/S-BSST889>).

32. Line 219: "Three mice were sacrificed during the experiment" Are these mice from those used in Figure 5 or are these extra mice?

- Please note that these are additional mice. We refer to these mice in lines 538-541 and Table 4 in the revised manuscript.

33. Figure 6E: What does this figure imply?

- This figure implies the significance of the differences between each animal group compared to the uninfected group; this is shown as a graph of *p*-values. We revised the figure and respective legend, hopefully clarifying the implied meaning.

34. Table 2: What is the phage used here?

- We apologize for the poor clarity regarding the same. By "phage", we meant to imply any tested phage, given the absence of differences between them. This has been clarified in Table 3 (previous Table 2), along with an added note.

35. Line 265-266; 268-269; 270-275: Because the bacteria were sensitive to either phage or Abx, so this is foreseeable that the bacteria is sensitive to combinational therapy. Therefore, the conclusion drawn by the authors may not be correct (also in the abstract).

➤ As mentioned above, phages do not specifically infect antibiotic-resistant bacteria. However, considering treatment, phage therapy is unessential when antibiotics exhibit sufficient efficacy. In cases when antibiotic-resistant strains are involved, phage therapy should be considered, and even then, probably combined with antibiotics. We do not anticipate the replacement of antibiotic therapy with phage therapy, but phages could be employed as an additional antibacterial tool. In the revised manuscript, we clarified this important point in lines 347-349 in the revised manuscript. Other lines described by the reviewer were removed from the revised manuscript.

36. Line 265-266: *In vivo* or *in vitro* model using *C. acnes* strain resistant to both phages and Abx should be used to confirm this statement.

➤ *In vitro* stimulation of antibiotic-resistant strain is described in Fig. 2. We addressed the issue for the *in vivo* model in comment #12

37. Line 309: "Table XX"???

➤ This has been corrected in the revised manuscript.

38. Figure 6: A phage control (uninfected mice using the cream) should also be investigated in the study to exclude the baseline level of immune activation by the phage

➤ We added this data as Fig 6D-E in the revised manuscript.

39. At last, the author should also expand their discussion on phage resistance, as this is another problem when using phage therapy. (In this study there are also phage-resistant bacteria)

➤ We agree with the reviewer that phage resistance is one of the most important issues concerning phage therapy. In general and very briefly, in contrast to antibiotics, when resistance emerges against phages, several steps can be employed to overcome it. First, phages can be evolutionarily mutated and selected, as in nature, to infect the resistant bacteria. Second, new phages can be isolated against resistant bacteria. Finally, phages can be suitably engineered. Moreover, although bacteria have numerous resistant systems, constantly being discovered every other day, we ultimately selected phages that efficiently infect the bacteria, regardless of these systems. We elaborated on these points in the discussion, in lines 372-388 in the revised manuscript

40. The discussion would flow more smoothly if it were more integrated. The current discussion section sounds more like a result section to me.

➤ We revised the discussion to afford better flow and readability.

41. Was an ELISA ever done to look at serum and cell-released levels of IL-1B? This may provide information about the general state of the mice.

➤ We thank the reviewer for this comment. Accordingly, ELISA was performed to measure serum levels of excreted IL1-β in each mouse group. This data is presented in Fig. 6G in the revised manuscript and discussed in lines 330-331 in the result section and in line 602 in the methods section of the revised manuscript.

42. Line 381: skin sample or skin swab?

➤ For clarity, we used the term "skin swabs" throughout the revised manuscript.

43. Line 432: 5 mM

➤ This error has been corrected.

44. Regarding sample collection and phage isolation: Was IRB approval obtained to collect the sample from patients? How was the sample collected? What are the characteristics of these patients? Are they Abx responsive? Did the saliva also collect from acnes pts? And more details are needed.

➤ In accordance with the reviewer's suggestion, we added the requested data in a new table (Table 1 in the revised manuscript) and the Materials and Methods section, *i.e.*, lines 444-473 of the revised manuscript. We also added a section in the discussion regarding the connection between patients' bacteria and phage isolation line 372-378 of the revised manuscript

45. When were the uninfected mice euthanized? How were they treated?

➤ Uninfected mice were euthanized after 3 days (before phage application) and on days 5 and 7 (after phage application). These were administered saline, along with sebum application. We included these details in Table 4 and in the Materials and Methods section in lines 537-541 of the revised manuscript.

46. Line 461: What liquid media? How was the skin incubated?

➤ We used Wilkins media. The skin was incubated for 24 h at room temperature. These details were added in lines: 603-607 of the revised manuscript.

47. Please move Figure 7 to Supp Fig.

➤ Figure 7 is presented as Supplemental Fig S6 in the revised manuscript.

48. Line 486-487: How was the phage primer designed?

- The primers were designed using Primer-Blast (<https://www.ncbi.nlm.nih.gov/tools/primer-blast>). This was added to the revised manuscript in lines 555-556.

49. Line 520: "RT-PCR For RNA isolation": RT-PCR was not used for RNA isolation

- We apologize for this error, which has been corrected in the revised manuscript.

50. Line 522: Where was Pearson analysis used?

- Pearson's correlation was performed in a previous version and omitted in the current manuscript version.

51. How was the sample size determined?

- The sample size was determined using the calculator: <https://clincalc.com/stats/samplesize.aspx>. This information has been included in line 531-533 of the revised manuscript

52. Line 526-527: cycling threshold is not "DCT" (should be "Ct")

- This error has been corrected in the revised manuscript.

53. Line 526-527: please spell out "2-DDC" and use the correct abbreviation.

- This error has been corrected in the revised manuscript

54. Line 534: Was this the approval for the animal experiment?

- Yes, this statement indicates the approval for animal experimentation. We rewrote it to improve clarity and provided another ethical approval for the clinical specimen isolation.

55. Table S4: Were these primers newly designed for the study? If so, please provide more details for each of them.

- The primers were obtained from previous reports, and references were added accordingly in Table S6.

56. Line 450-453: But there are five samples on day 5 (Fig 6D). Also, where are the data from day 7?

- We apologize for this error. The day number was corrected throughout the manuscript.

Figures and Figure Legends:

57. Table 2: which phage?

- By "phage," we meant to imply each phage, given the lack of differences between them. See also our answer to comment 34. This point was clarified in Table 3 (previously 2) and the respective legend.

58. Table 2 legend: "Some isolates were verified as non-*C. acnes* and therefore some numbers are missing" what does this mean?

- Sample numbers were determined during appropriate sampling. However, some bacterial isolates were not identified as *C. acnes* by 16S analysis and were removed from the study. Nevertheless, these were included in Table S4 in the revised manuscript.

59. Figure 1: The figure is difficult to see.

- A higher resolution figure was provided.

60. All figures: Low-resolution images are not acceptable.

- All low-resolution figures were replaced with high-resolution images.

61. Figures 2C and F are not necessary; or consider moving them to the Supp Fig.

- These figures were moved to the Supplementary section, with 2C presented as S2F and 2F as S3C in the revised manuscript.

62. Figure 3: Did image capturing and pathology study also take place on day 8? (as shown in Supp Table 3)

- We apologize for this error. The numbering in Supplemental Table 3 was erroneous and was rectified.

63. Line 805-806: Again, the macrophage and neutrophils cannot be distinguished from the figure.

- This has been rectified in the figure and respective legend.

64. Figure 6A-C: The group name should be added to the figure.

- As requested, this was rectified, and the order was altered to match the sequence in the text.

65. Figure 6A-C: Gating strategies should be added in Supp Figure

➤ The gating strategies were added as Supplemental Fig. S8B in the revised manuscript.

66. Figure 6F: I can't see from the figure which group was used as a normalizer to calculate relative expression? (The value of the normalizer seems to differ in each group?)

➤ We corrected Figure 6F; the uninfected group was used as a normalizer.

67. More details are needed in all the figure legends (both main Figure and Supp Figure)

➤ Figure legends were revised throughout the manuscript, with additional details supplied. We hope that these revisions improve their clarity and implications.

68. Supp Table 3: The author should take all the pictures from the same angle, light source, background, etc. For example, images on day 8 from control animals #15 and #16 are totally different than the rest of the picture, which made us difficult to interpret the data. Other examples such as Day 7 from treated animal #1; Day 8 from treated animal #16; as well as others!

➤ Images were obtained from the same location and in the same room with similar lighting using the same camera. However, owing to the movement of mice, it was impossible to achieve identical angles and backgrounds. Unfortunately, improving the images at this point would warrant the additional euthanization of 34 new animals, and we do not believe that can be justified. However, we added a scale bar to the image to aid in better interpretation.

69. Supp Table 3: Also, the author should provide a scale bar (*e.g.* ruler with a scale bar next to the lesion site) when taking the picture.

➤ As mentioned in the previous comment, we added a scale bar for each image using the measured size of the lesions.

70. Supp Fig 5: How were the macrophages gated?

➤ Gating strategies were added to Supplemental Fig. 8B in the revised manuscript.

71. Supp Fig 5: It is interesting that there is no significant difference between days 3 and 5 (in the FD3 group) when the level is apparently high on day 5. Was this comparison done?

➤ We added these comparisons in Supplemental Fig. 8A and the text in lines 320-322

72. Supp Fig 5 legend and Supp Fig 5: In the legend the author stated "monocytes", however in the figure it showed "macrophages"

➤ In the revised manuscript legend, "monocytes" was corrected to macrophages.

73. All the tables and figures need to be reproduced to provide clear contents (as well as a clear and informative figure legend)

➤ All tables, figures, and their legends were revised and reproduced.

Other minor points for improvement:

74. An abstract should be typed as a single paragraph.

➤ In the revised manuscript, the abstract is presented as a single paragraph.

75. The author please recheck the manuscript carefully for spelling mistakes. For example:

Line 341: medhanisms or mechanisms? Crrrossig or crossing?

➤ The revised manuscript underwent another round of professional English editing in Wiley Editing Services (<https://wileyeditingservices.com/en/>, see attached certificate)

Line 427: Phylogeneteic or phylogenetic?

➤ "Phylogenetic". This was rectified in the manuscript.

Line 299: type in italics

➤ As requested, the text is presented in italics.

76. Line 242: Figure 1F should be Figure 6F.

➤ This error has been rectified in the revised manuscript.

Reviewer #2 (Remarks to the Author):

This work shows that use of phages in a mouse model of Cutibacterium acne brings a benefit, although not a notably large benefit. The work is well designed and executed, although its subsequent write-up is less attractive. The manuscript needs to be carefully and thoroughly edited to achieve correct scientific English. There are numerous mistakes in spelling, punctuation, grammar, and incorrect use of capital letters. The sentence in line 450 no verb.

➤ Given that our team comprises non-native English speakers, we had the manuscript undergo professional English language editing, which was insufficient. The revised manuscript was once again edited by a native English editor, Wiley Editing Services (<https://wileyeditingservices.com/en/>), who also supplied a certificate for it. We truly hope that it is much better now.

The term 'sacrifice' which generally means 'slaughtering an animal or person as an offering to a deity' is an unacceptable euphemism for "kill"; "sacrifice" should be replaced throughout with a less euphemistic term such as kill or euthanase.

➤ We replaced the term "sacrifice" with "euthanize" throughout the manuscript.

The main issue with this work seems to be around the animal model. The paper would be improved by:

(i) Discussion justifying the suitability of the model, and the route of delivery of the bacteria. Why is it injected, and not, for example, applied to scarified skin? Given that the model required two injections of a relatively large volume of bacteria at high concentration, is the model really representative of what happens in acne?

➤ Herein, we aimed to simulate acne vulgaris as feasible. Acne is a human disease and does not have clinical correlates in animals. Therefore, establishing an animal model simulating acne pathogenicity can be challenging. We thoroughly reviewed the literature and selected a mouse model described by Kolar *et al.*⁸, which mimics human disease better than other reported animal models^{8,9}.

This model involved the intradermal administration of bacteria, similar to other accepted acne animal models, as well as the application of artificial sebum to best simulate the environment of the human hair follicle. Thus, in order to induce the lesions, we had to apply high concentrations of bacteria. Note, though, that at the endpoint, when assessing CFU counts, the number of bacteria was much lower.

Nevertheless, our major goal in this work was to demonstrate the feasibility of treating skin infections induced by *C. acnes* using phages. Despite the limitations, this goal was achieved and opened the way for further experiments toward acne vulgaris phage-based treatments. We discussed this in lines 414-429 of the revised manuscript.

(ii) Improved description of how the phage was applied. Neither Materials and Methods nor Figure 3 gives enough detail about how the phage was applied. Was the gel simply smeared over the skin lesion? If so, what volume of gel? Was any intervention used to keep the gel on the lesion and stop the mice from rubbing it off? If not, for how long was the gel in place on each lesion?

➤ We added all requested data on the phage application in lines 534-541 in the revised manuscript. Briefly, 0.5 ml of Carbopol gel containing $10^9 \frac{\text{PFU}}{\text{ml}}$ FD3 phage was applied daily to each lesion. We could not prevent the mice from rubbing it off; however, based on our observation, the gel was still fully present after 3 h and partially detectable after 6 h post-administration. We aimed to establish a treatment protocol that

may be relevant in clinical settings. A unique feature of phages is auto-dosing, as phages undergo replication on their target bacterium.

References

1. Yu, Y., Champer, J. & Kim, J. Analysis of the surface, secreted, and intracellular proteome of *Propionibacterium acnes*. *EuPA Open Proteomics* **9**, 1–7 (2015).
2. Miescher, S. Antimicrobial and autolytic systems of dairy *propionibacteria*. (1999).
3. Lam, H. Y. P. *et al.* Therapeutic Effect of a Newly Isolated Lytic Bacteriophage against Multi-Drug-Resistant *Cutibacterium acnes* Infection in Mice. *Int. J. Mol. Sci.* **22**, 7031 (2021).
4. Sheffer-Levi, S. *et al.* Antibiotic Susceptibility of *Cutibacterium acnes* Strains Isolated from Sheffer-Levi, S., Rimon, A., Lerer, V., Shlomov, T., Copenhagen-Glazer, S., Rakov, C., Zeiter, T., Nir-Paz, R., Hazan, R., and Molcho-Pessach, V. (2020). Antibiotic Susceptibility of *Cu*. *Acta Derm. Venereol.* (2020).
5. Liu, J. *et al.* The diversity and host interactions of *Propionibacterium acnes* bacteriophages on human skin. *ISME J.* **9**, 2078–2093 (2015).
6. Marinelli, L. J. *et al.* *Propionibacterium acnes* bacteriophages display limited genetic diversity and broad killing activity against bacterial skin isolates. *MBio* **3**, (2012).
7. Shuta, T. *et al.* Pan-Genome and Comparative Genome Analyses of *Propionibacterium acnes* Reveal Its Genomic Diversity in the Healthy and Diseased Human Skin Microbiome. *MBio* **4**, e00003-13 (2022).
8. Kolar, S. L. *et al.* *Propionibacterium acnes*-induced immunopathology correlates with health and disease association. *JCI insight* **4**, (2019).
9. Jang, Y. H., Lee, K. C., Lee, S.-J., Kim, D. W. & Lee, W. J. HR-1 Mice: A New Inflammatory Acne Mouse Model. *ad* **27**, 257–264 (2015).

REVIEWER COMMENTS

Reviewer #1 (Remarks to the Author):

The authors have attended to many of the points raised in my first review (and also to the comments from another reviewer); however, some issues remain regarding the animal experiment, along with minor irregularities with English expression (see below):

1. A major problem with the animal experiment is that the authors did not present the results of the control group which is essential (such as in figure 4) to data interpretation. As intradermal injection may also induce some histological changes, lacking the control group made us impossible to understand whether the histological changes were indeed caused by bacterial colonization and not due to injections.
2. Another problem: "...as the control group here was also treated with phage gel" (this is only a treatment-control, which excludes side effects of the phage), what about mice being injected but not being treated with phage gels? (this should be the control group) This group should be presented in all experiments.
3. The author mentioned that the gel was still fully present after 3 h and partially detectable after 6 h post-administration; does it mean it was not absorbed well into the skin? Did the author perform any tests to determine that the phage was indeed being absorbed? If the gel were still present on the skin, did the author remove them after a period of time?
4. In Fig S5, please provide the uncropped agar images. Also, if the phages were dropped on the skin fragment, how can the author make sure that the phages droplet on the skin can not be washed out by the medium? Also, details regarding this experiment may need to be clearly explained, such as the thickness and sizes of the skin? Is there any reference supporting this experimental design? There are multiple experiments and devices used to detect drug penetration, is this experiment any better than those?
5. The use of English was much improved in the revised manuscript, but some mistakes were still spotted and the author may wish to check carefully again before their final manuscript can be published. Also, the author please check again regarding their data presentation, such as the labeling of intercepts.
6. S1 legends: typo error of "x40K"?
7. S4 legends: "Parameters of the phage..." what does it mean?
8. S4 legends: What is the composition of the control tube?
9. Line 86: the word "phage" does not need to be in italics

10. Line 108: ...*"Pahexavirus of the Siphoviridae family"* word in italics
11. L111: *"Siphoviridae"*, word in italics
12. L134: Although authors already mentioned that the phages belong to Siphoviridae, according to phylogenetic analysis, did TEM analysis reveal the same results? Please also state the results in this section.
13. Line 132-133, 162-167, 215-226, and throughout the manuscript.: "SD 17.4 nm" should be SD=17.4 nm, as well as others mentioned in the manuscript
14. Line 141, 4 should be four
15. Line 144-145: "Within the 32 phage-sensitive stains, 11 of them were resistant to at least one antibiotic..." and "... to all five tested antibiotics" Sounds better
16. Line 162 to 167: similarly, "SD=..." should be used
17. Line 159 and 173: ICR mice or BALB/c mice?
18. Line 215: "5.2 mm, SD 0.6 AU" ... How come the units are different? What are they referring to? If the authors are not mentioning the same thing, please state it clearly (very confusing to the readers!) In my opinion, SD should be listed first because you are talking about the scores; then the diameter as additional information. For example, ...with a score of 1 ± 0.6 AU (diameter of ? mm)
19. Line 270: "... subsequently subjected to histopathological assessment to establish cellular infiltration of lesions" Incorrect use of English.
20. Line 271 and Figure 4 legends: "control mice" mean untreated or placebo-treated mice, did the author imply "infected mice"?
21. Line 279: "based on the pathological report" please delete this sentence
22. How about the histopathological examination of the control mice (uninfected, non-phage-treated mice)?
23. Line 321, "infected" does not need to be in italics
24. Line 327: (ns Fig. 6F). Typo error?
25. Line 342: delete the word "value"
26. Line 390: italics
27. L481: CLSI guidelines
28. L485: How were the skin swabs or saliva samples collected? Were these samples also collected from pts where the bacteria were isolated?
29. L536 as well as others: should be "with or without"; "at a concentration of"
30. L537: "The groups were separated to prevent the transfer of phages between them." What does this mean? Wasn't different groups separately housed?

31. Typo: uninfected
32. The author can just simply mention the group as control, phage-control, infected, and treated group
33. L561, 608, etc: incorrect use of English
34. Fig 3: control or infected
35. Fig 3I-L: The mice were injected on day 1, so why not data from day 1 and 2 also presented here?
36. Fig 4: when were these mice sacrificed? This should be presented in the legend as the author sacrifice mice at multiple time points.
37. Fig 5D: y-ints: total number? total number of how many fields? Or number per field? uninfected control should be listed first.
38. Fig 6D-E and S8: what about data from day 3 in the treated group|?
39. Fig 6F and G: control group should be presented first.
40. Fig S8B: both CD45+CD11b- and CD45+CD11b+ cells were gated to sub-gate CD64+ cells and Ly6G+ cells, was this gating strategy correct?
41. Supp Table S4 Legend: this is understandable that the movement of mice made imaging difficult. As the authors mentioned in the response notes, please also add the statement in the Table S4 legend.

Please find our point-to-point answers to the reviewers' comments on our manuscript, entitled "*Toward phage therapy for acne vulgaris: Topical application in a mouse model of Cutibacterium acnes-induced acne-like lesions*" NCOMMS-22-20 by Rimon *et al.*

We want to thank the reviewer for the constructive comments that improved our manuscript significantly, and we agree with all the comments provided and revised the manuscript accordingly.

The major changes we made were the addition of the control of the saline injection, showing that *Cutibacterium acnes*-induced acne-like lesions resulted from bacterial infection and not from injection *per se*. In addition, we revised the skin-penetration phage assay

Please see below our point-to-point revisions (in blue):

REVIEWER COMMENTS

Reviewer #1 (Remarks to the Author):

1. A major problem with the animal experiment is that the authors did not present the results of the control group which is essential (such as in figure 4) to data interpretation. As intradermal injection may also induce some histological changes, lacking the control group made us impossible to understand whether the histological changes were indeed caused by bacterial colonization and not due to injections.

➤ We thank the reviewer for this comment and agree that testing the effect of the injection itself is an important control. To this end, we injected mice with saline and performed all the described assays for this control. For the histology, Fig. 4 was revised accordingly, and we also referred to this in the text (lines 285-287 in the results section and lines 442-443 in the discussion).

2. Another problem: "...as the control group here was also treated with phage gel" (this is only a treatment-control, which excludes side effects of the phage), what about mice being injected but not being treated with phage gels? (this should be the control group) This group should be presented in all experiments.

➤ As described above in comment #1 for the histology, the saline injected control group was not treated with phages and was added to each relevant assay. It appears now in Figs 3-6 and the relevant text.

3. The author mentioned that the gel was still fully present after 3 h and partially detectable after 6 h post-administration; does it mean it was not absorbed well into the skin? Did the author perform any tests to determine that the phage was indeed being absorbed? If the gel were still present on the skin, did the author remove them after a period of time?

➤ We assessed phage presence inside the lesion after disinfecting the outer layer of the lesion (Fig. 3B). This experiment demonstrated that phages had been absorbed into the lesion. To further test whether the phages' penetration into the skin was not due to the injection tract, we performed an improved *ex-vivo* assay where phages in Carbopol gel were applied on uninjured non-infected, nor injected skin, followed by an assessment of phage quantification in a segregated compartment. We found that phages indeed penetrate the skin, as shown in Fig S5. For more details, see the next comment.

Regarding gel removal, the gel was not removed because it was applied as a simulation of clinical application, where the gel is left to absorb without intervention. This is now clarified in line 564 of the revised manuscript.

4. In Fig S5, please provide the uncropped agar images. Also, if the phages were dropped on the skin fragment, how can the author make sure that the phages droplet on the skin can not be washed out by the medium? Also, details regarding this experiment may need to be clearly explained, such as the thickness and sizes of the skin? Is there any reference supporting this experimental design? There are multiple experiments and devices used to detect drug penetration, is this experiment any better than those?

- We thank the reviewer for this comment. We supply further evidence for the ability of phages to penetrate the skin. In Kumar *et. al* from 2012¹, it was shown that *E. coli* targeting phages penetrated a full-thickness mice back skin. We used a similar model to show that our anti-*C. acnes* phages are also able to penetrate. To this end, we used the model described by Rohrschneider *et al*² where skin pieces were placed in donor compartments of trans-wells. A solution of phages mixed with the water soluble Allora Red dye was applied on the uninjured skin, followed by assessment at several time points of the dye and titer of phages in the recipient compartments (Fig. S5). This experiment demonstrated that the phages penetrate the skin while the dye, representing the solvent, does not. These experiments are described in Fig. S5, results (Lines 192-199), discussion (Lines 434-436), and methods (Lines 637-648) sections of the revised manuscript.

5. The use of English was much improved in the revised manuscript, but some mistakes were still spotted and the author may wish to check carefully again before their final manuscript can be published. Also, the author please check again regarding their data presentation, such as the labeling of intercepts.

- Although we used two English editing services, we found and corrected some more typos and errors, including in some axis labels in the figures. We apologize and hope the language of the whole manuscript is acceptable now.

6. S1 legends: typo error of “x40K”?

- This typo was corrected according to the target journal's guidelines.

7. S4 legends: "Parameters of the phage..." what does it mean?

- The “Parameters of the phage...” means their stability and efficacy. To clarify this, we altered Fig. S4 legend to:
- **“FigS4. Phage selection for in-vivo experiments** phage stability and efficacy were assessed to select one phage, or a phage cocktail for in-vivo *Cutibacterium acnes*-induced acne-like lesions model. (A) Stability in the gel. The eight phages’ activity was assessed following incubation in Carbopol gel for 30 days, with samples taken on days 1, 5, 10, 15, 20, and 30. Each day, PFU was calculated from a gel tube and a matching control tube containing a phage in Wilkins medium. (B) Comparison of a single phage activity using FDI, FD3, and PAVL45 vs. a cocktail containing all three phages on *C. acnes* strain #27. Based on these experiments, FD3 was selected as a single phage for the in-vivo experiments.”

These parameters are also described in lines 161-174 of the revised manuscript.

8. S4 legends: What is the composition of the control tube?

- The composition of the control tube is the phage in Wilkins medium. We clarified this in the legend of Fig. S4: "PFU was calculated from a gel tube and a matching control tube containing a phage in Wilkins medium." This was also clarified in line 162 of the revised manuscript.

9. Line 86: the word "phage" does not need to be in italics

- Corrected.

10. Line 108: ..." Pahexavirus of the Siphoviridae family" word in italics

- Corrected.

11. L111: "Siphoviridae", word in italics

- Corrected.

12. L134: Although authors already mentioned that the phages belong to Siphoviridae, according to phylogenetic analysis, did TEM analysis reveal the same results? Please also state the results in this section.

- Yes, the TEM analysis confirmed that the phages' shape matches that of previously described *Siphoviridae*. This was added in lines 130-131 of the revised manuscript.

13. Line 132-133, 162-167, 215-226, and throughout the manuscript.: "SD 17.4 nm" should be SD=17.4 nm, as well as others mentioned in the manuscript

- SD was corrected throughout the revised manuscript.

14. Line 141, 4 should be four

- Corrected.

15. Line 144-145: "Within the 32 phage-sensitive stains, 11 of them were resistant to at least one antibiotic..." and "... to all five tested antibiotics" Sounds better.

- Corrected.

16. Line 162 to 167: similarly, "SD=..." should be used

- SD was corrected throughout the revised manuscript.

17. Line 159 and 173: ICR mice or BALB/c mice?

- ICR mice. This was corrected in the revised manuscript.

18. Line 215: "5.2 mm, SD 0.6 AU"... How come the units are different? What are they referring to?

If the authors are not mentioning the same thing, please state it clearly (very confusing to the readers!) In my opinion, SD should be listed first because you are talking about the scores; then the diameter as additional information. For example, ...with a score of 1 ± 0.6 AU (diameter of ? mm)

➤ Corrected accordingly (lines 222-239 of the revised manuscript).

19. Line 270: "... subsequently subjected to histopathological assessment to establish cellular infiltration of lesions" Incorrect use of English.

➤ This sentence was revised to: " biopsies were obtained from all groups and cellular infiltration of lesions was assessed by histopathology." (lines 284-285 of the revised manuscript).

20. Line 271 and Figure 4 legends: "control mice" mean untreated or placebo-treated mice, did the author imply "infected mice"?

➤ Mice group names were changed throughout the manuscript as described below in comment #32

21. Line 279: "based on the pathological report" please delete this sentence

➤ Deleted.

22. How about the histopathological examination of the control mice (uninfected, non-phage-treated mice)?

➤ As described above in comment #1, this control group was added throughout the manuscript, and this specific point is addressed in lines 285-287 of the revised manuscript.

23. Line 321, "infected" does not need to be in italics

➤ Corrected.

24. Line 327: (ns Fig. 6F). Typo error?

➤ Corrected.

Line 342: delete the word "value"

➤ Deleted.

26. Line 390: italics

➤ Corrected.

27. L481: CLSI guidelines

➤ Corrected.

28. L485: How were the skin swabs or saliva samples collected? Were these samples also collected from pts where the bacteria were isolated?

- Phages were isolated from the same skin swabs described in the bacterial isolation section and from saliva samples – collected from the same patients using a sterile swab. This was clarified in lines 512-513.

29. L536 as well as others: should be "with or without"; "at a concentration of"

- Corrected throughout the revised manuscript.

30. L537: "The groups were separated to prevent the transfer of phages between them." What does this mean? Wasn't different groups separately housed?

- Yes, different groups were housed separately to prevent phage transfer between them. We clarified this point in the revised manuscript in lines (565-566).

31. Typo: uninfected

- Corrected.

32. The author can just simply mention the group as control, phage-control, infected, and treated group

- We changed the groups' nomenclature throughout the revised manuscript to: a. sham-injected b. phage control, c. infected-untreated, and d. infected and phage-treated (treated).

33. L561, 608, etc: incorrect use of English

- Corrected.

34. Fig 3: control or infected

- The group names were revised as described in comment #32.

35. Fig 3I-L: The mice were injected on day 1, so why not data from days 1 and 2 also be presented here?

- Injections were performed on days 1 and 2. On these days, no change was observed. Only on day 3 were the mice randomly allocated into groups for the different treatments and scoring was performed. Thus, we show the results from day 3. This was explained in line 563, and in the legend of Fig. 3 of the revised manuscript.

36. Fig 4: when were these mice sacrificed? This should be presented in the legend as the author sacrifice mice at multiple time points.

- Mice were euthanized on day 10. We supplied euthanasia time information in the legend of Fig. 4 of the revised manuscript.

37. Fig 5D: y-ints: total number? total number of how many fields? Or number per field? uninfected control should be listed first.

➤ This is the number per field. We corrected this description of the y-axis.

38. Fig 6D-E and S8: what about data from day 3 in the treated group ?

➤ On day 3 which is the peak of disease the group allocation took place. Therefore, mice sacrificed at this time point did not receive treatment. Day 3 is the setpoint for both of the groups, treated and untreated. This was elaborated in line 563, and in the legend of Fig. 6 and Fig. S8.

39. Fig 6F and G: control group should be presented first.

➤ Corrected.

40. Fig S8B: both CD45+CD11b- and CD45+CD11b+ cells were gated to sub-gate CD64+ cells and Ly6G+ cells, was this gating strategy correct?

➤ We thank the reviewer for pointing this out and indeed we found this gating strategy was inaccurate. This was corrected, we added references for each gating strategy and elaborated on this in lines 626-627, and in the Legend of Fig. S8 of the revised manuscript.

➤ Nevertheless, the data interpretation was only mildly altered, as can be observed in the figure below, in which we present histograms for levels of CD11b in the cell population presented in the former Fig. S8, according to the incorrect gating strategy: LY6G+ cells (Letter Fig. 1A) and CD64+ cells (Letter Fig. 1B)

Letter Figure 1. Histogram of CD11b levels in both populations gated in the previous version of Fig. S8. (A) LY6G+ population. (B) CD64+ population.

41. Supp Table S4 Legend: this is understandable that the movement of mice made imaging difficult. As the authors mentioned in the response notes, please also add the statement in the Table S4 legend.

- This statement was added to the revised legend of table S4.

References:

1. Kumar, S., Sahdev, P., Perumal, O. & Tummala, H. Identification of a novel skin penetration enhancement peptide by phage display peptide library screening. *Mol. Pharm.* **9**, 1320–1330 (2012).
2. Rohrschneider, M. *et al.* Evaluation of the transwell system for characterization of dissolution behavior of inhalation drugs: effects of membrane and surfactant. *Mol. Pharm.* **12**, 2618–2624 (2015).
3. Supe, S. & Takudage, P. Methods for evaluating penetration of drug into the skin: A review. *Ski. Res. Technol.* **27**, 299–308 (2021).

REVIEWERS' COMMENTS

Reviewer #1 (Remarks to the Author):

The authors have attended to all the points raised in my review. Seeing this work has been done, I think it might be reasonable to publish, after addressing the minor points below:

1. Line 104-105: "Based on the absence of these phage sequences in bacterial genomes..." What experiments did the author perform to prove this statement?
2. Line 160: "...remarkable ability..." the author may state what remarkable ability made this strain the most suitable model to use?
3. Line 181: "...into two groups (n=17)" should be "into two groups (n=17 per group)"
4. Line 188: "...groups, sham injected, treated with phage and vehicle"; Does the author mean "All mice groups, including sham control and phage control group did not exhibit..."
5. Line 230: "and days 3-5" should be "and from day 3 to day 5"
6. Line 242: "... it reduced to 0 by day 7..." should be "which reduced to 0 by day 7"
7. The author analyzed the level of IL-17, CXCL2, TNF- α , and IL-6; therefore the author may wish to give a brief discussion of these cytokines because, except IL-6, these are not direct inflammatory cytokines. For example, what are their relationship to neutrophil or macrophages?
8. Line 380: should be "induced by"
9. Line 437: should be "on the first day"
10. Line 463: "could be observed" sounds better
11. Line 538: did the isolated phage sequence upload to the NCBI databank? (All genomic data should be uploaded to the databank before publishing the manuscript) If so, the accession number should be provided in this section.
12. Line 554: "...34 ICR eight-week-old albino mice" should be "34 eight-week-old, albino strain, ICR mice". Another piece of information lacking here is the gender of the mice.
13. Line 596: "...xylene, and 100, 95, 80, and 70% ethanol washed three times with PBS,"; should be "xylene, 100%, 95%, 80%, and 70% ethanol before washing three times with PBS"
14. Line 611: "100 μ g/mL of..." the author may unite the typing of all units in the manuscript: "mL or ml"
15. Line 639: "from mice"... also ICR mice? at what age? Male or female?
16. Line 861: should be "The skin is within..."

17. Line 887: should be “Quantification of the neutrophils number in flow cytometric analysis. Data are presented as mean \pm standard deviation (SD).”

Dear Reviewer of Nature Communications,

Please find our point-to-point answers to the reviewers' comments regarding our manuscript, now entitled "*Topical phage therapy in a mouse model of Cutibacterium acnes-induced acne-like lesions*" NCOMMS-22-20 by Rimón *et al.*

We want to thank the reviewer for the constructive comments which improved our manuscript's discussion and language. We agree with all the comments. The major addition in this version based on the reviewer's request is the addition of a paragraph in the discussion regarding the various cytokines assessed in this work and clarification of several points. In this version, all changes are in track changes, and row lines are numbered accordingly.

Please see below our point-to-answers (in blue):

REVIEWER COMMENTS

The authors have attended to all the points raised in my review. Seeing this work has been done, I think it might be reasonable to publish, after addressing the minor points below:

1. Line 104-105: "Based on the absence of these phage sequences in bacterial genomes..." What experiments did the author perform to prove this statement?

- To examine whether a phage is lysogenic or lytic, we performed a BLAST analysis and assessed the presence of the whole phage genome sequence in bacterial genomes. A phage is considered lysogenic if most blast hits are within bacterial genomes. On the other hand, a phage is considered lytic if most of the blast hits are phages. This is described in lines 104-108 of the Results section of the revised manuscript.

2. Line 160: "...remarkable ability..." the author may state what remarkable ability made this strain the most suitable model to use?

- This bacterial strain was arbitrarily selected between several strains tested for this model. It is now clarified in lines 162 of the Results section and 584-585 of the Methods section of the revised manuscript. As remarkable is not an objective description, we removed it from our manuscript.

3. Line 181: "...into two groups (n=17)" should be "into two groups (n=17 per group)"

- Corrected.

4. Line 188: "...groups, sham injected, treated with phage and vehicle"; Does the author mean "All mice groups, including sham control and phage control group did not exhibit..."

- Corrected.

5. Line 230: "and days 3-5" should be "and from day 3 to day 5"

- Corrected.

6. Line 242: "... it reduced to 0 by day 7..." should be "which reduced to 0 by day 7"

- Corrected.

7. The author analyzed the level of IL-17, CXCL2, TNF- α , and IL-6; therefore the author may wish to

give a brief discussion of these cytokines because, except IL-6, these are not direct inflammatory cytokines. For example, what are their relationship to neutrophil or macrophages?

- We discussed this in the revised manuscript in lines 462-476 as follows: Neutrophil migration to the lesion was reduced, as well as *IL-6*, *IL-17*, *CXCL2*, *TNF α* gene expression in the phage-treated group in comparison to the infected-untreated group. These specific cytokines were chosen due to their involvement in pro-inflammatory processes, as well as macrophage and neutrophil function¹⁻⁵. *IL-6* is a pro-inflammatory cytokine secreted by non-hematopoietic cells as well as by different immune cells¹ playing a role in the differentiation of T-helper 17 cells (Th17)¹. Another cytokine we assessed, *IL-17* is secreted by Th17 and $\gamma\delta$ T-cells^{3,6}, a subset of T-cells important for the initiation of the inflammatory response⁶. This cytokine mediates differentiation and activation of macrophages², as well as recruitment and activation of neutrophils in the tissue³. Part of its effect is facilitated through the induction of *CXC* chemokines³, such as *CXCL2* from non-hematopoietic cells. *CXCL2* attracts neutrophils to the tissue⁴ and enables their passage through epithelial tight junctions⁴. Finally, *TNF α* , produced by activated macrophages, is responsible for a diverse range of cell signaling cascades, leading to necrosis or apoptosis⁵. These inflammatory processes contribute to the pathogenesis of acne vulgaris⁷.

The following references were added accordingly:

1. Scheller, J., Chalaris, A., Schmidt-Arras, D. & Rose-John, S. The pro-and anti-inflammatory properties of the cytokine interleukin-6. *Biochim. Biophys. Acta (BBA)-Molecular Cell Res.* **1813**, 878–888 (2011).
2. de la Paz Sánchez-Martínez, M. *et al.* IL-17-differentiated macrophages secrete pro-inflammatory cytokines in response to oxidized low-density lipoprotein. *Lipids Health Dis.* **16**, 1–9 (2017).
3. Li, X., Bechara, R., Zhao, J., McGeachy, M. J. & Gaffen, S. L. IL-17 receptor-based signaling and implications for disease. *Nat. Immunol.* **20**, 1594–1602 (2019).
4. Metzemaekers, M., Gouwy, M. & Proost, P. Neutrophil chemoattractant receptors in health and disease: double-edged swords. *Cell. Mol. Immunol.* **17**, 433–450 (2020).
5. Idriss, H. T. & Naismith, J. H. TNF α and the TNF receptor superfamily: Structure-function relationship (s). *Microsc. Res. Tech.* **50**, 184–195 (2000).
6. Roark, C. L., Simonian, P. L., Fontenot, A. P., Born, W. K. & O'Brien, R. L. $\gamma\delta$ T cells: an important source of IL-17. *Curr. Opin. Immunol.* **20**, 353–357 (2008).
7. Cong, T.-X. *et al.* From pathogenesis of acne vulgaris to anti-acne agents. *Arch. Dermatol. Res.* **311**, 337–349 (2019).

8. Line 380: should be "induced by"

- Corrected.

9. Line 437: should be "on the first day"

➤ Corrected.

10. Line 463: "could be observed" sounds better

➤ Corrected.

11. Line 538: did the isolated phage sequence upload to the NCBI databank? (All genomic data should be uploaded to the databank before publishing the manuscript) If so, the accession number should be provided in this section.

➤ All of the isolated phage sequences were uploaded to the NCBI Genebank (<https://www.ncbi.nlm.nih.gov/genbank/>), and association numbers are supplied (Table 2). This is now described in lines 564-565 of the revised manuscript.

12. Line 554: "...34 ICR eight-week-old albino mice" should be "34 eight-week-old, albino strain, ICR mice". Another piece of information lacking here is the gender of the mice.

➤ Corrected. Mice's gender was added.

13. Line 596: "...xylene, and 100, 95, 80, and 70% ethanol washed three times with PBS,"; should be "xylene, 100%, 95%, 80%, and 70% ethanol before washing three times with PBS"

➤ Corrected.

14. Line 611: "100 µg/mL of..." the author may unite the typing of all units in the manuscript: "mL or ml"

➤ Corrected throughout the revised manuscript

15. Line 639: "from mice"... also ICR mice? at what age? Male or female?

➤ 8-week-old, female ICR, albino strain mice. This is now described in lines 679, 680 of the revised manuscript.

16. Line 861: should be "The skin is within..."

➤ Corrected.

17. Line 887: should be "Quantification of the neutrophils number in flow cytometric analysis. Data are presented as mean ± standard deviation (SD)."

➤ Corrected. We have moved the second sentence, "Data are presented as mean ± standard deviation (SD)" to the end of the legend, because it is relevant for Fig. 6 d, f, g.